# NSm is a critical determinant for bunyavirus transmission between vertebrate and mosquito hosts

Selim Terhzaz ⬤ , David Kerrigan, Floriane Almire, Agnieszka M. Szemiel ⬤ ,
Joseph Hughes ⬤ , Jean-Philippe Parvy, Massimo Palmarini ⬤ , Alain Kohl,
Xiaohong Shi ⬤ ✉ & Emilie Pondeville ⬤ ✉

*Bunyavirales* is a very large order including viruses infecting a variety of taxonomic groups such as arthropods, vertebrates, plants, and protozoa. Some bunyaviruses are transmitted between vertebrate hosts by blood-sucking arthropods and cause major diseases in humans and animals. It is not understood why only some bunyaviruses have evolved the capacity to be transmitted by arthropod vectors. Here we show that only vector-borne bunyaviruses express a non-structural protein, NSm, whose function has so far remained largely elusive. Using as experimental system Bunyamwera virus (BUNV) and its invertebrate host, *Aedes aegypti*, we show that NSm is dispensable for viral replication in mosquito cells in vitro but is absolutely required for successful infection in the female mosquito following a blood meal. More specifically, NSm is required for cell-to-cell spread and egress from the mosquito midgut, a known barrier to viral infection. Notably, the requirement for NSm is specific to the midgut; bypassing this barrier by experimental intrathoracic infection of the mosquito eliminates the necessity of NSm for virus spread in other tissues, including the salivary glands. Overall, we unveiled a key evolutionary process that allows the transmission of vector-borne bunyaviruses between arthropod and vertebrate hosts.

The *Bunyavirales* order is a large group of viruses that infect many different species of arthropods, vertebrates, plants, and protists. Some bunyaviruses have evolved to cycle between blood sucking arthropods and vertebrates, and they represent approximately half of the currently known arthropod-borne viruses (arboviruses). Arboviral diseases pose significant threats to human and animal health worldwide, leading to substantial economic burdens[1,2]. With no vaccines available for most of these diseases, and limited effective vector control measures[3,4], the growing impact of climate change will expand the reach of arboviral threats, amplifying disease control challenges[5–7]. Prioritizing research to understand factors facilitating arbovirus transmission and develop integrated control strategies is crucial for

public health, as exemplified by the recent launch of the World Health Organization's Global Arbovirus Initiative[8]. Among arboviruses, vector-borne bunyaviruses (arbo-bunyaviruses) pose one of the greatest public health risks. For example, Crimean-Congo haemorrhagic fever virus (CCHFV, transmitted by ticks) and Rift Valley fever virus (RVFV, transmitted by mosquitoes), are included in the list of infectious diseases prioritised by the WHO for requiring further research and development[9].

Although bunyaviruses have diverse host range and pathogenicity, they share key features including possessing a segmented and single-stranded RNA genome encoding a small number of proteins, a similar virion structure, viral replication in the cytoplasm, and

MRC-University of Glasgow Centre for Virus Research, Glasgow G61 1QH Scotland, UK. [2]Present address: Departments of Tropical Disease Biology and Vector Biology, Liverpool School of Tropical Medicine, Liverpool L3 5QA, UK. ✉e-mail: Xiaohong.Shi@glasgow.ac.uk; Emilie.Pondeville@glasgow.ac.uk

assembly and budding at membranes of the Golgi complex[10–12]. They possess a membrane envelope coated by glycoproteins and their genome, generally consisting of three segments, small (S), medium (M), and large (L), is wrapped up by the nucleocapsid protein N and interacts with the viral polymerase. The S segment encodes the N protein and in many cases a non-structural protein NSs, which modulates the interferon response of the mammalian host[12]. The M segment encodes the glycoprotein precursor (GPC) that is co-translationally cleaved by host peptidases to yield the mature viral glycoproteins Gn and Gc, while the large (L) segment encodes the RNA-dependent RNA polymerase L. In some viruses from certain bunyavirus families (*Peribunyaviridae*, *Nairoviridae*, *Phenuiviridae* and *Tospoviridae*), the M segment also encodes a non-structural protein NSm[12]. While the role of the structural proteins and NSs is well known across bunyavirus families[12], no clear function for the NSm protein has been identified[12,13], with the exception of the thrip-transmitted plant virus Tomato spotted wilt virus (TSWV, *Tospoviridae* family), where NSm acts as a movement protein involved in cell-to-cell transport of viral ribonucleoproteins[14,15].

In Bunyamwera virus (BUNV, *Peribunyaviridae*), the prototype of bunyaviruses, NSm is a 15 kDa type II integral membrane protein containing five domains, three hydrophobic domains (I, III, and V) and two non-hydrophobic domains (II and IV)[16,17]. The N-terminal domain I serves as NSm signal peptide and is cleaved during GPC processing. The C-terminal domain V serves as signal peptide for downstream Gc but remains integral to NSm[17]. In BUNV-infected mammalian cells, NSm is targeted to the Golgi complex and co-localizes with viral glycoproteins Gn and Gc. Since virus assembly and budding at the Golgi is one of the characteristics of bunyaviruses, it was suggested that NSm may be involved in virus assembly and morphogenesis[16,18]. However, we have shown that, except for the N-terminal domain I, which serves as the NSm signal peptide and is required for BUNV morphogenesis, BUNV is able to tolerate deletions of all other domains (domains II to V), indicating that NSm is largely dispensable for virus replication in mammalian cells[16,17]. This was further confirmed with several other bunyaviruses (Maguari virus, *Peribunyaviridae*[19]; Oropouche virus, *Peribunyaviridae*[20]; Schmallenberg virus, *Peribunyaviridae*[21]; RVFV, *Phenuiviridae*[22,23]; CCHFV, *Nairoviridae*[24]).

Here we show that NSm is only present in bunyaviruses transmitted by arthropods. Since arboviruses circulate between a vertebrate host (or plants in some cases) and an arthropod vector, and no clear function of NSm has been described in vertebrates, we hypothesised that NSm has a specific role during the viral life cycle in the arthropod host. We investigated the role of NSm using both in vitro experiments in cell cultures and in vivo studies in the female mosquito. We provide several lines of evidence demonstrating that NSm is required for virus cell-to-cell spread in the midgut, and dissemination to secondary tissues in the mosquito. Our study reveals that NSm was acquired during the evolution of vector-borne bunyaviruses, and it is a critical determinant allowing virus transmission between vertebrate and arthropod hosts.

## Results

### NSm is only present in bunyaviruses transmitted by arthropods
To gain insights into the function of NSm, we performed an *in-silico* analysis of the presence or absence of NSm across viruses from the *Bunyavirales* order. Interestingly, this revealed a strong correlation between the presence of NSm and bunyaviruses vectored by arthropods (Fig. 1). Indeed, NSm is present in arbo-bunyaviruses only and all bunyaviruses transmitted by arthropods encode an NSm protein, with a few exceptions in the *Phenuiviridae* family. All bunyaviruses which are not arboviruses do not encode NSm. As NSm is present in bunyaviruses belonging to only a few viral genera, and which are dispersed across phylogenetically distinct families[12], the "law of parsimony" supports a convergent evolution with

independent and repeated acquisitions of NSm in bunyaviruses. To further infer the origin and evolution of NSm across bunyaviruses, we then blasted the BUNV NSm sequence to retrieve all homologous sequences and performed a phylogenetic analysis (Fig. S1a). While NSm from most of *Peribunyaviridae* could be retrieved and aligned (distance mean: 0.8539, minimum distance: 0.1502, maximum distance: 0.9396), NSm from other families, *i.e.*, *Nairoviridae*, *Phenuiviridae* and *Tospoviridae*, could not be retrieved due to an absence of protein sequence similarity. This is also confirmed by the lack of similarity between the structure prediction of NSm from different families (BUNV, RVFV, CCHFV and TSWV), as shown Fig. S1b. This supports a single ancestral acquisition of NSm within the *Peribunyaviridae*, followed by divergent evolution in each virus, and independent acquisitions of NSm – likely different genes/sequences now all called NSm- by the different virus families.

### NSm is dispensable for virus replication in mammalian and arthropod cells
We have previously shown that BUNV NSm is largely dispensable for virus replication in mammalian BHK-21 cells[16,17]. Using a BUNV NSm deletion mutant (BUN-ΔNSm, Fig. S2a), we confirmed that NSm is dispensable for virus replication in different mammalian cells (Fig. S2b). Since our analysis highlighted that NSm is only present in bunyaviruses transmitted by arthropods, we next investigated whether NSm was required for replication in arthropod cells. BUNV is transmitted by mosquitoes, we therefore assessed the impact of its deletion on the viral replication kinetics in different mosquito cell lines (*Ae. albopictus* larval C6/36 and U4.4 cell lines and *Ae. aegypti* embryonic Aag2 cell line). Across all the cell lines tested (Fig. S2b), the growth curves of BUNV wild type (BUNV-wt) and BUN-ΔNSm were not significantly different, showing that NSm is not required for viral replication in mosquito cells.

### NSm is required for BUNV infection in vivo in the female mosquito
We next tested the requirement of NSm for successful infection of the mosquito vector. To investigate this possibility, we fed *Ae. aegypti* females with blood infected with either BUNV-wt or BUN-ΔNSm (Fig. S3a) and assessed the infection (bodies) and dissemination (heads) titres at 3-, 6-, and 16-days post-blood meal (dpbm) (Fig. 2a). In the body of females infected with BUNV-wt, viral titres increased between 3, 6 and 16 dpbm showing successful infection and replication of BUNV-wt over time (Fig. 2b). In the head of BUNV-wt infected mosquitoes, we detected viral titres from 6 dpbm and later, consistent with the time required for viral dissemination from the midgut to distal tissues to occur. Although we detected BUN-ΔNSm in the body at 3 dpbm only, albeit to lower titres compared to BUNV-wt, we could not detect BUN-ΔNSm in the head of the mosquitoes at all three time points (Fig. 2b), suggesting that no dissemination occurred in mosquitoes infected with the NSm deletion mutant.

To further delineate the importance of the NSm protein at the tissue level, we fed mosquitoes with a BUNV-wt or BUN-ΔNSm infectious blood meal and determined viral titres at 3 and 15 dpbm specifically in the midgut and salivary gland tissues, as well as in the rest of the body (Fig. 2c). As expected, we observed increased viral titres from 3 to 15 dpbm in both the midgut and body, with effective dissemination to the salivary glands between 3 and 15 dpbm in BUNV-wt infected mosquitoes. However, we could detect only very low levels of BUN-ΔNSm in midguts at 3 dpbm and not at all at 15 days post-infection. Furthermore, we confirmed that no dissemination occurred at 3 and 15 dpbm since only one body sample was infected at 15 dpbm and no salivary glands were infected. Taken together, these results revealed the crucial role of BUNV NSm in vivo as the deletion of NSm alone was sufficient to nearly abolish midgut infection and dissemination in the *Ae. aegypti* mosquito.

## NSm is not required for dissemination and transmission when bypassing the midgut

Since the midgut is the first tissue to be infected by an arbovirus before disseminating to other tissues and considering that BUN-ΔNSm was not able to infect the gut efficiently, we wondered whether the NSm requirement was specific to the midgut or generally required for infection of mosquito tissues. To assess this, we intrathoracically inoculated female mosquitoes with either BUNV-wt or BUN-ΔNSm to overcome the midgut barrier (Fig. 3a). We sampled several whole mosquitoes minutes after injection to ensure that females from each group were inoculated with an equivalent number of viral particles (Fig. S3b). We collected midguts and salivary glands individually at 3- and 9-days post-injection (dpi) and subsequently measured virus titres (Fig. 3b). At both 3 and 9 dpi, we found that the viral titres in midguts from BUN-ΔNSm injected mosquitoes were significantly lower

compared to the BUNV-wt injected mosquitoes. However, we did not observe differences in the salivary glands at 3 and 9 dpi, indicating that both BUNV-wt and BUN-ΔNSm equally disseminated to and infected the salivary glands, when bypassing the midgut barrier through intra-thoracic inoculation. To unambiguously validate this observation and determine if the mosquitoes were able to transmit BUN-ΔNSm virus, we repeated the injection experiments and measured the viral titres in individual salivary glands and in the mosquito saliva at 7 dpi as a proxy of transmission. Again, we confirmed that there was no significant difference in viral titres in the salivary glands (Fig. 3c) with comparable production of infectious saliva in both BUNV-wt and BUN-ΔNSm injected mosquitoes (Fig. 3d).

We also analysed the level of virus infection by assessing the expression of the viral N protein in both midgut and salivary gland tissues. In agreement with the titration result, we observed that N

| Virus family | Virus genus | Virus species example | Host | NSm | Arbovirus | Arthropod vector |
|---|---|---|---|---|---|---|
| *Arenaviridae* | All | *Lymphocytic choriomeningitis virus* | Vertebrates | No | No | NA |
| *Cruliviridae* | All | *European shore crab virus* | Arthropods | No | No | NA |
| *Fimoviridae* | All | *Fig mosaic virus* | Plants | No | No | NA |
| *Hantaviridae* | All | *Puumala virus* | Vertebrates | No | No | NA |
| *Leishbuviridae* | All | *Leptomonas moramango virus* | Protists | No | No | NA |
| *Mypoviridae* | All | *Húběi myriapoda virus* | Arthropods | No | No | NA |
| *Nairoviridae* | *Norwavirus* | *Běijí nairovirus* | Arthropods | No | No | NA |
| | *Ocetevirus* | *Red goblin roach virus* | Arthropods | No | No | NA |
| | *Orthonairovirus* | *Crimean-Congo haemorrhagic fever virus* | Vertebrates | Yes | Yes | Ticks |
| | *Sabavirus* | *South Bay virus* | Arthropods | No | No | NA |
| | *Shaspivirus* | *Shāyáng spider virus* | Arthropods | No | No | NA |
| | *Striwavirus* | *Sānxiá water strider virus* | Arthropods | No | No | NA |
| | *Xinspivirus* | *Xīnzhōu spider virus* | Arthropods | No | No | NA |
| *Peribunyaviridae* | *Herbevirus* | *Herbet virus* | Arthropods | No | No | NA |
| | *Khurdivirus* | *Khurdun virus* | Vertebrates | No | No | NA |
| | *Lakivirus* | *Lakivirus* | Arthropods | No | No | NA |
| | *Lambavirus* | *Lambavirus* | Vertebrates | No | No | NA |
| | *Orthobunyavirus* | *Bunyamwera virus* | Vertebrates | Yes | Yes | Mosquitoes, Midges |
| | *Pacuvirus* | *Pacui virus* | Vertebrates | Yes | Yes | Sandflies |
| | *Shangavirus* | *Shangavirus* | Arthropods | Yes | ? | |
| *Phasmaviridae* | All | *Orthophasmavirus aedis* | Arthropods | No | No | NA |
| *Phenuiviridae* | *Bandavirus* | *Dabie bandavirus* | Vertebrates | No | Yes | Ticks |
| | *Banyangvirus* | *Guertu virus* | Vertebrates | No | Yes | Ticks |
| | *Beidivirus* | *Húběi diptera virus* | Arthropods | No | No | NA |
| | *Citricivirus* | *Aphis citricidus bunyavirus* | Arthropods | No | No | NA |
| | *Coguvirus* | *Citrus concave gum-associated virus* | Plants | No | No | NA |
| | *Entovirus* | *Entoleuca phenui-like virus* | Fungi | No | No | NA |
| | *Goukovirus* | *Yíchāng insect virus* | Arthropods | No | No | NA |
| | *Horwuvirus* | *Wǔhàn horsefly virus* | Arthropods | No | No | NA |
| | *Hudivirus* | *Húběi diptera virus* | Arthropods | No | No | NA |
| | *Hudovirus* | *Húběi lepidoptera virus* | Arthropods | No | No | NA |
| | *Ixovirus* | *Blacklegged tick virus* | Arthropods | No | No | NA |
| | *Laulavirus* | *Wardell virus* | Fungi | No | No | NA |
| | *Lentinuvirus* | *Lentinuvirus lentinulae* | Plants | No | No | NA |
| | *Mechlorovirus* | *Melon chlorotic spot virus* | Plants | No | No | NA |
| | *Mobuvirus* | *Ñarangue virus* | Arthropods | No | No | NA |
| | *Phasivirus* | *Badu virus* | Arthropods | No | No | NA |
| | *Phlebovirus* | *Rift Valley Fever virus* | Vertebrates | Yes | Yes | Mosquitoes, Sandflies |
| | *Pidchovirus* | *Coleopteran phenui-related virus* | Arthropods | No | No | NA |
| | *Rubodvirus* | *Apple rubbery wood virus* | Plants | No | No | NA |
| | *Tanzavirus* | *Dar es Salaam virus* | Vertebrates | No | No | NA |
| | *Tenuivirus* | *Rice stripe virus* | Plants | No | Yes | Planthoppers, Leafhoppers |
| | *Uukuvirus* | *Uukuniemi Virus* | Vertebrates | No | Yes | Ticks |
| | *Wenrivirus* | *Mourilyan virus* | Arthropods | No | No | NA |
| *Tospoviridae* | All | *Tomato spotted wilt virus* | Plants | Yes | Yes | Thrips |
| *Wupedeviridae* | All | *Wǔhàn millipede virus* | Arthropods | No | No | NA |

**Fig. 1 | Correlation between bunyaviruses vectored by arthropods and the presence of NSm.** Analysis of the presence (highlighted in blue) or absence (highlighted in red) of NSm across all families and genera within the *Bunyavirales* order. Arboviruses are highlighted in blue and bunyaviruses which are not arboviruses are highlighted in red. Primary host taxonomic group is shown in addition to the vector host for arboviruses. *Shangavirus* encodes a putative NSm, but biology and host specificity of this virus is unknown. Phleboviruses (e.g., Rift Valley Fever virus) also encode P78, which is processed from the glycoprotein precursor initiated from the first start codon of the M segment and covers both NSm and Gn coding region. Data were collected from the International Committee on Taxonomy of Viruses (https://ictv.global/taxonomy).

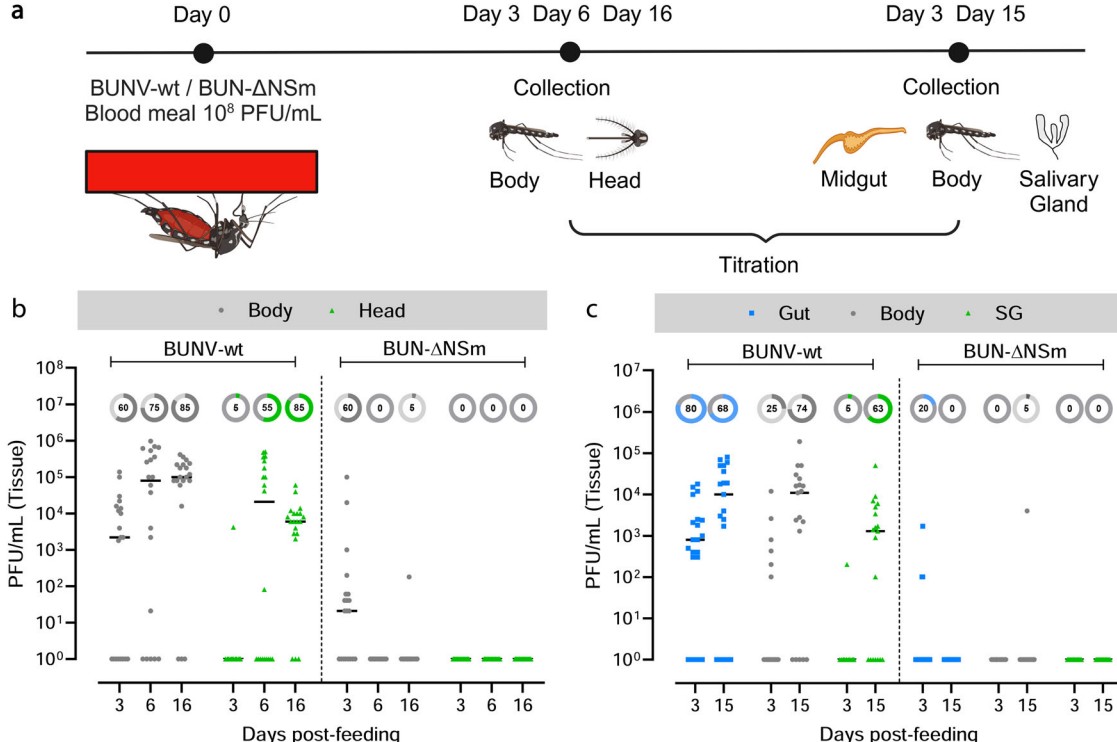

**Fig. 2 | Effect of NSm deletion on infection and dissemination following oral feeding. a** Experimental scheme for the analysis of BUNV-wt and BUN-ΔNSm infection dynamics (Created in BioRender. Terhzaz, S. (2024) https://BioRender.com/o44x482). *Ae. aegypti* were fed with a blood meal containing $4 \times 10^8$ PFU/mL of BUNV-wt or BUN-ΔNSm. Virus titres were measured by plaque assay on BHK-21 cells. **b** BUNV-wt or BUN-ΔNSm titres in bodies and heads at 3-, 6- and 16-days

post-blood meal (dpbm). **c** BUNV-wt or BUN-ΔNSm titres in guts, bodies, and salivary glands (SG) at 3 and 15 dpbm. Titres are displayed as PFU/mL for each tissue with individual samples displayed (*n* = 20 per condition). All samples are presented in the graph. Lines indicate median values and the circled numbers represent the percentage of infected samples for each condition. Source data are provided as a Source Data file.

staining in midguts was strongly reduced in BUN-ΔNSm inoculated mosquitoes compared to BUNV-wt ones. At 3 dpi, N was strongly detected in the muscle fibres and the tracheal system surrounding the midgut epithelium of BUNV-wt infected mosquitoes (Fig. 3e and S3c; control in Fig. S4a). At 9 dpi, the virus infected the midgut with the presence of large clusters of epithelial cells expressing N protein (Fig. 3e, arrow). In BUN-ΔNSm injected mosquitoes, midgut N staining was less prominent and, interestingly, we never detected foci of infection even at 9 dpi (Fig. 3e). In contrast to the midgut, N staining was very similar in salivary glands of BUNV-wt and BUN-ΔNSm injected mosquitoes at both 3 and 9 dpi with increased level of N protein over time (Fig. 3f; control in Fig. S4a). These data show that both viruses can enter acinar cells and replicate to high levels within this tissue if mosquitoes are infected intrathoracically instead of receiving an infected blood meal. Dissemination to other secondary cells/tissues was also confirmed in hemocytes, the circulating immune cells, in which N staining was similar between the two viruses at both 3 and 9 dpi (Fig. 3g).

Taken together, our data showed that the impairment of BUN-ΔNSm is specific to the midgut. When bypassing the midgut, NSm is not required for infection and dissemination to secondary tissues, resulting in successful transmission potential.

**BUN-ΔNSm infects midgut cells but does not form infection foci**
We have established that BUN-ΔNSm infection and dissemination is blocked at the level of the mosquito midgut. Therefore, we next sought to identify the stage at which BUN-ΔNSm infection is arrested by assessing the early dynamics of BUNV-wt and BUN-ΔNSm using virus titration, immunofluorescence and RT-qPCR (Fig. 4a). We fed *Ae. aegypti* females with a blood meal containing BUNV-wt or BUN-ΔNSm and then determined virus titres in whole females at 0 h and at 1, 2, 3

and 6 dpbm (Fig. 4b). We found no differences in virus titres between the two groups at 0 h. As expected, BUNV-wt titres increased steadily over time from 1 to 6 dpbm. Remarkably, we could not detect any infectious virus in BUN-ΔNSm fed females at all time points tested, suggesting that BUN-ΔNSm failed to efficiently infect the midgut at an early stage.

To further assess these data, we quantified the BUNV segment S RNA levels by RT-qPCR in the midgut at 3, 24, 48 and 72 h pbm (Fig. 4c). Interestingly, in contrast to virus titres, the levels of viral segment S levels were similar for both viruses at 3 and 24 h, which suggests that viral RNA from initially ingested virions persists in the midgut for 24 h post-blood meal, likely in the gut lumen while the blood is not yet fully digested. Between 24 and 48 h, an eclipse phase was observed as BUNV S levels dropped for both BUNV-wt and BUN-ΔNSm, albeit to significantly lower levels for BUN-ΔNSm compared to BUNV-wt. However, BUNV-wt levels significantly increased between 48 and 72 h pbm, exhibiting the post-eclipse pattern of RNA replication, whereas BUN-ΔNSm levels continued to decline at 72 h pbm. These results confirmed that BUN-ΔNSm failed to efficiently infect the entire midgut tissue within 3 days post-infection as shown previously (Fig. 2c).

Next, we analysed the infection pattern in the midgut epithelium early after oral infection at 24, 48 and 72 h pbm (Fig. 4d). At 24 h pbm, we detected N staining in individual cells throughout the midgut epithelium in both BUNV-wt and BUN-ΔNSm infected mosquitoes, indicating that both viruses can efficiently enter midgut epithelial cells soon after feeding and suggesting that BUN-ΔNSm infection is impaired after initial cell entry in the midgut. The dynamics of BUNV-wt and BUN-ΔNSm infections were distinctly different over time. At 48 h pbm, we detected N staining in numerous clusters of cells, or foci of infection, in the midgut of BUNV-wt infected mosquitoes, whereas staining was still confined to a few isolated midgut cells in BUN-ΔNSm

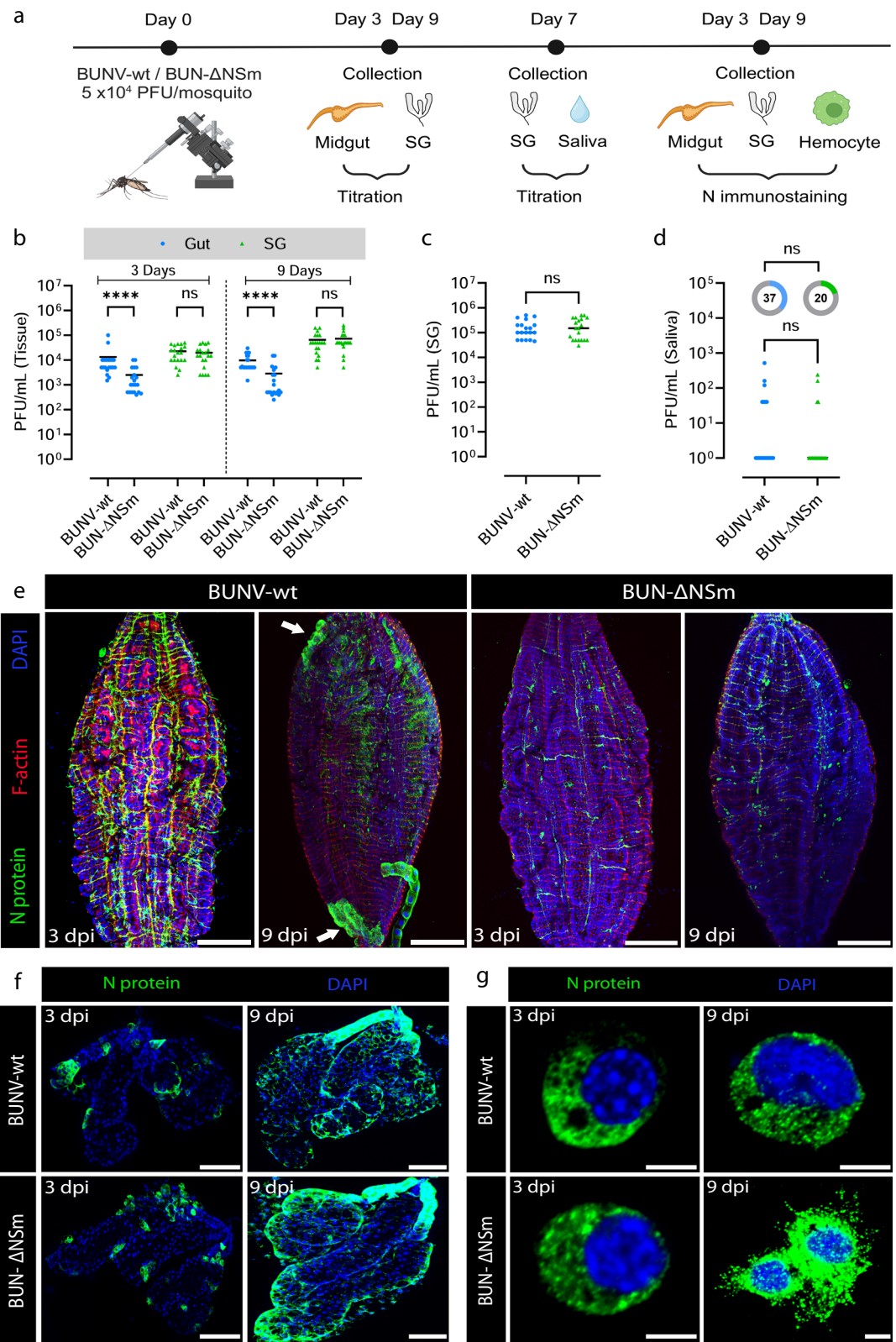

infected mosquitoes. At 72 h pbm, the difference between the two viruses was markedly increased. In BUNV-wt infected mosquitoes, we detected N antigen in large foci that spread throughout the midgut, compared to those observed at 48 h pbm. Higher magnification of an individual focus of infection showed a large cluster of cells (>100 cells) with intense N staining in the cytoplasm (Fig. 4d, circle). By contrast, only a small number of individual midgut cells were positive

for the viral N antigen at 72 h pbm after BUN-ΔNSm infection, showing that BUN-ΔNSm stays confined in the initially infected cells and does not spread to neighbouring cells to form large foci. Altogether, our data indicated that the NSm deletion mutant was able to enter and infect midgut cells, but without NSm, the virus failed to spread from infected to uninfected cells and thus propagate in the midgut epithelium.

**Fig. 3 | Effect of NSm deletion on infection, dissemination and transmission following injection in *Ae. aegypti*. a** Experimental scheme for the analysis of BUNV-wt and BUN-ΔNSm infection dynamics (Created in BioRender. Terhzaz, S. (2024) https://BioRender.com/s57y407). **b** Virus titres in midguts and salivary glands (SG) at 3 and 9 days after BUNV-wt or BUN-ΔNSm exposure by intrathoracic injection (5 × 10^4 PFU/mosquito). Virus titres were measured by plaque assay on BHK-21 cells. Titres are displayed as PFU/mL for each tissue with individual samples displayed (*n* = 20 per condition). Statistical significance shown on the graph was obtained using a two-way ANOVA followed by a Tukey's multiple comparisons test. ns, not significant, *p* value > 0.5; ****p* value < 0.0001. Adult females were inoculated intrathoracically with 5 × 10^4 PFU/mosquito of BUNV-wt (*n* = 19) or BUN-ΔNSm (*n* = 20) and virus titres of individual salivary glands (**c**) and saliva (**d**) quantified at 7 dpi by plaque assay on BHK-21 cells. Viral titres were analysed by a two-tailed Mann-Whitney test and the infection prevalence was analysed with a Chisquare test. ns = not significant. All the samples are presented in the graph. Lines indicate median values and the circled numbers represent the percentage of infected mosquitoes. Infection prevalence for the saliva were not different (*p* = 0.25). Midguts (**e**) and salivary glands (**f**) were dissected at 3 and 9 dpi and stained with anti-N recognizing the viral nucleocapsid N protein (green), Phalloidin Texas Red to visualize F-actin filaments (red) and with DAPI to visualize nuclei (blue). Images are merged z-stacks. Scale bars are 150 μm for (**e**) and 100 μm for (**f**). **g** Circulating hemocyte cells perfused from mosquitoes at 3 and 9 dpi and stained with DAPI to visualize nuclei (blue), and with anti-N (green). Scale bars are 10 μm. Source data are provided as a Source Data file.

## In vivo expression of NSm in trans partially rescues BUN-ΔNSm infection in the midgut

To confirm that the impairment of BUN-ΔNSm in the midgut was due to the absence of the NSm protein, we next tested if in vivo transfection of an expression vector for BUNV NSm could rescue mosquito infection following oral delivery of BUN-ΔNSm (Fig. 5a). We microinjected mosquitoes with a plasmid containing the BUNV NSm gene under the control of the polyubiquitin gene promoter (pPUb-NSm-V5). At 6 dpi, we examined V5-tagged NSm protein expression in the midgut of pPUb-NSm-V5 injected mosquitoes and found many cells labelled by the V5 antibody (Fig. 5b–e; control in Fig. S4b), suggesting that the NSm protein was efficiently expressed in the mosquito midgut. Higher magnification images showed V5 staining in the cytoplasm as well as at the plasma membrane of midgut epithelial cells (Fig. 5e, arrow). We also fed mosquitoes with a blood meal containing BUN-ΔNSm and assessed N expression and viral titres at 3 dpbm. Concomitant with the ectopic expression of the NSm protein, the infection prevalence and viral titres in pPUb-NSm-V5 injected mosquitoes were significantly higher than in control mosquitoes (Fig. 5f). Additionally, we found a significantly higher number of cells with N staining along with the reestablishment of infection foci in the midgut of pPUb-NSm injected mosquitoes compared to control mosquitoes (Fig. 5g–i). Infection foci were smaller than the ones observed in the gut of BUNV-wt infected mosquitoes (Fig. 4d), consistent with NSm-V5 not being expressed in every midgut cell (Fig. 5b–e). Taken together, our data show that the ectopic expression of the NSm gene significantly rescued BUN-ΔNSm infection in *Ae. aegypti*, demonstrating that BUNV NSm is a crucial viral determinant necessary for the successful spread of infection in the mosquito midgut.

## Higher doses of BUN-ΔNSm does not enhance virus spread or dissemination

Midgut barriers encountered by arboviruses can be virus dose-dependent or -independent, depending on the underpinning mechanism, *e.g.*, if these barriers are associated with the inability of the virus to overcome an antiviral response or with cell/tissue surface structures that the virus cannot efficiently traverse[25]. To obtain further insights into the mechanism by which NSm is required to overcome the midgut infection and escape barriers, we next assessed if higher infectious doses of BUN-ΔNSm virus could impact midgut cell-to-cell spread of infection and dissemination (Fig. 6a). We fed mosquitoes with a blood meal containing 2 x 10^7, 7.7 x 10^7 or 2 x 10^8 PFU/mL of either BUNV-wt or BUN-ΔNSm, and quantified virus titres in whole females at 3 dpbm (Fig. 6b). In BUNV-wt fed mosquitoes, viral titres and infection rates were high, but there was no significant difference for the doses tested. In BUN-ΔNSm infected mosquitoes, we found again a dramatic reduction of viral titres and infection rates. Although the number of infected mosquitoes rose from 5 to 20–25% with increasing virus titres in the inoculum, no significant difference in viral titres was found among females infected with different doses of BUN-ΔNSm virus. We also performed immunostaining analysis using the N antibody to determine how a dose increase would affect the number of infected cells and foci formation in the midgut. At 3 dpbm, we detected BUNV N antigen in just a few isolated cells over the entire midgut epithelium, and the number of N protein expressing cells was significantly higher in the guts of mosquitoes fed with a 10-fold higher dose of BUN-ΔNSm virus (Fig. 6c). However, we were never able to visualize large infection foci as consistently seen in BUNV-wt fed mosquitoes, showing that even at higher doses, BUN-ΔNSm virus can infect a greater number of midgut epithelial cells immediately after the blood meal, but the mutant virus is unable to spread to neighbouring cells to form infection foci.

We next investigated if the increase of the number of infected midgut cells observed at the higher dose of BUN-ΔNSm could potentially lead to virus dissemination within the mosquito. We fed mosquitoes with a blood meal containing either 2 x 10^7 or 2 x 10^8 PFU/mL of BUN-ΔNSm, and quantified virus titres in the midguts and bodies at 3 and 6 dpbm (Fig. 6d). We observed higher infection prevalence and virus titres in the midgut of mosquitoes infected with higher titres of BUN-ΔNSm in the blood meal. However, this did not result in dissemination from the midgut to the rest of the body. Finally, we also found that the number of infected cells in the midgut, assessed by N staining, did not increase between day 2, 3 and 6 post-infection (Fig. 6e), demonstrating that the initial infection by BUN-ΔNSm does not spread over time regardless of viral input dose.

Altogether, our findings showed that ingestion of higher doses of NSm deletion mutant virus increases the number of infected cells, but this is not sufficient for the virus to spread in the midgut or to disseminate from the midgut, suggesting that both the midgut infection and escape barriers encountered by BUN-ΔNSm are dose-independent.

## NSm is expressed on the cell surface of infected cells and at the periphery of foci in BUNV infected midguts

In mammalian cells, it was previously shown that the BUNV NSm protein co-localized with viral glycoproteins Gn and Gc to the Golgi complex[16]. In BUNV-infected mosquito cells, we found some levels of co-localisation of Gc and NSm in the cytoplasm (Fig. 7a, top right panel, insert image), but to a lesser extent compared to mammalian cells. We next performed Gc and NSm immunostaining on non-permeabilized insect cells infected with BUNV-wt. Interestingly, we found that both NSm and Gc proteins were localized at the cell surface (Fig. 7a, bottom right panel, insert image). This result was confirmed in mosquito cells infected with the recombinant BUNV-NSmV5. Indeed, the V5-tagged NSm protein was abundantly expressed at the cell periphery, while Gc was localised in punctate structures in the cell as well as at the cell membrane (Fig. 7b). Remarkably, surface staining also revealed punctate staining of NSm in thin finger-like extensions protruding from infected mosquito cells (Fig. 7b, arrow).

Next, we performed in vivo immunocytochemistry studies to determine the NSm localisation in midgut epithelial cells of BUNV-wt infected mosquitoes. After an infectious blood meal, BUNV enters the midgut epithelium, and the infection spreads from primary infected cells to adjacent cells and then further progresses to form large

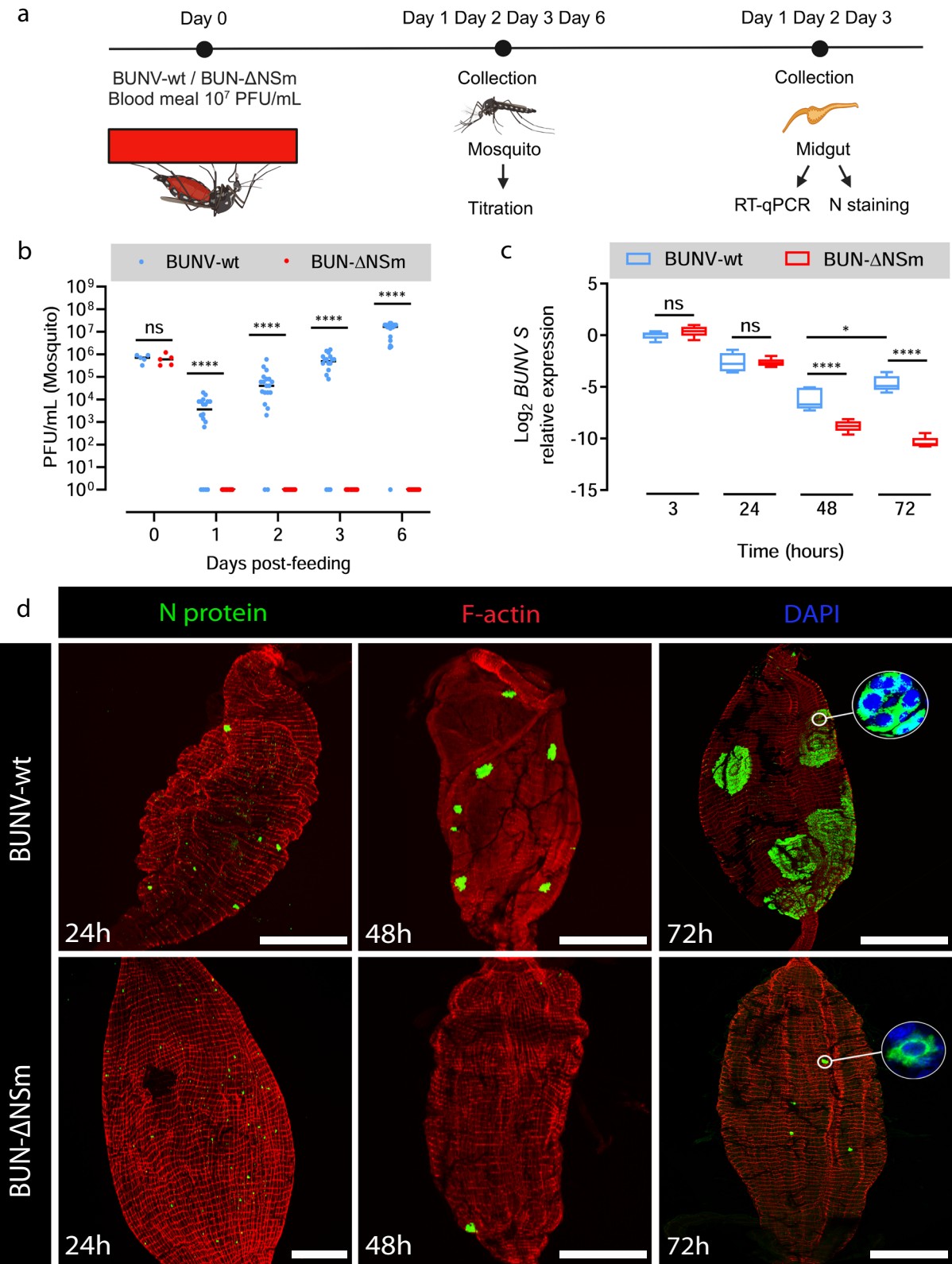

clusters of infected cells, ultimately leading to virus dissemination to secondary target tissues. Over time, these infection foci became enlarged and were visualized in the entire midgut epithelium (Fig. 4d), confirming previous observations with different arboviruses[26,27]. Interestingly, unlike the N (Figs. 4d and 6c) and Gc proteins (Fig. 7d–f), which accumulate in the cytoplasm and remains at high levels in every cell of the infection foci, the NSm protein was detected mainly at the periphery of foci (Fig. 7c–f, Fig. S5a, b), showing higher levels of expression in newly infected cells in these clusters. NSm and Gc proteins co-localized at membrane regions of infected cells, and particularly at their tricellular contacts – where the corners of three cells meet (Fig. 7g, yellow arrows). Taken together, we show that both Gc and NSm are detected at the cell surface. However, NSm is predominantly detected at the periphery of infection foci, suggesting that, unlike

**Fig. 4 | Infection pattern of BUNV-wt and BUN-ΔNSm in whole female *Ae. aegypti* and midguts. a** Experimental scheme for the analysis of BUNV-wt and BUN-ΔNSm infection dynamics (Created in BioRender. Terhzaz, S. (2024) https://BioRender.com/h11o587). **b** BUNV-wt or BUN-ΔNSm titres in mosquitoes at 0 h and 1, 2, 3 and 6 dpbm fed with a blood meal containing $2 \times 10^7$ PFU/mL of virus. Virus titres were measured by plaque assay on BHK-21 cells and are displayed as PFU/mL for each animal with individual samples displayed ($n = 5$ per condition for 0 h, $n = 16$–20 per condition for 1, 2, 3 and 6 dpbm). Statistical significance shown on the graph was obtained using a two-way ANOVA followed by a Tukey's multiple comparisons test. ns, not significant, $p$ value > 0.5; ****$p$ value < 0.0001. **c** BUNV S RNA levels in the midgut at 3, 24, 48 and 72 h pbm were quantified by RT-qPCR. Normalised expression for each sample was obtained as described, normalised to the S7 ribosomal gene and as relative values to that of the control group (BUNV-wt 3 h, RQ geomean set to 1). Log2-transformed RQ values were plotted ($n = 5$ pools of 4 midguts per group and time point). Box plots display the RQ minimum, first quartile, median, third quartile, and maximum. Statistical testing by two-way ANOVA followed by a Tukey's multiple comparisons test. *$p = 0.0225$; ****$p$ value < 0.0001. **d** Mosquitoes were fed with blood containing $7 \times 10^7$ PFU/mL of BUNV-wt or BUN-ΔNSm. Midguts were dissected at 24, 48 and 72 h pbm and stained with anti-N (green), F-actin filaments (red) and with DAPI to visualize nuclei (blue). Representative images are merged z-stacks. High-magnification images of the circled area showing individual or group of cells. Scale bars are 150 μm. Source data are provided as a Source Data file.

other viral proteins, NSm is expressed early and transiently during cell infection.

## Discussion

The *Bunyavirales* order is a large taxonomic group of viruses which include important human, veterinary, and plant pathogens. This order encompasses 13 phylogenetically distinct families whose viruses infect a variety of arthropods, vertebrates, plants, fungi and protists[12,28,29]. While all bunyaviruses possess a similar genome structure, viruses from some genera scattered in certain families also encode the nonstructural NSm protein.

Our analysis revealed that NSm is present in vector-borne bunyaviruses only. Several studies using bunyaviruses from different families have shown that NSm is largely dispensable for virus replication in mammalian cells[12,16,17,19–24]. We showed that NSm is also not required for BUNV replication in mosquito cells in vitro, as described previously for Oropouche virus, another bunyavirus[20], suggesting that NSm is not necessary for arbo-bunyavirus replication regardless of the host. However, we demonstrated that mosquitoes infected with BUN-ΔNSm had significantly lower midgut infection rates and dissemination in secondary tissues was abolished compared to mosquitoes infected with the wild type virus.

To successfully infect an arthropod vector, an arbovirus overcomes different midgut barriers. Initially, in the first few hours after the infectious blood meal, the virus invades and replicates in a few epithelial midgut cells. The virus then spreads from infected cells to uninfected ones within the midgut epithelium, before the virus finally escapes from the midgut[25]. We showed that NSm is specifically required for virus cell-to-cell spread in the mosquito midgut and is therefore an essential determinant for successful infection and dissemination in the arthropod vector. When we infected mosquitoes by intrathoracic inoculation, thus bypassing the midgut barrier, both the NSm deletion mutant and wild type virus infected the same peripheral tissues and cell types. For example, both viruses efficiently infected hemocytes, as described previously for other arboviruses[30], as well as tracheae, the respiratory system of insects, and the muscle fibres surrounding the midgut[31,32]. Similarly, BUN-ΔNSm could infect and replicate in salivary gland acinar cells and be released into the saliva. However, while BUNV-wt could infect the midgut epithelium from the basal side and form infection foci, BUN-ΔNSm was unable to do so. The titres reached by BUN-ΔNSm in the midgut were significantly reduced compared to those reached by wild type virus, with the absence of infection foci. Nonetheless, they were still higher than those reached after an infectious blood meal, as the mutant virus successfully infected tracheae and muscles surrounding the midgut.

After a blood meal infection, we found that BUN-ΔNSm can enter and replicate in a few cells scattered throughout the midgut. However, while BUNV-wt infection efficiently spread over time to form large infection foci within the gut epithelium, BUN-ΔNSm remained confined to individual midgut cells and did not spread to adjacent cells. Importantly, we rescued this mutant phenotype in mosquitoes by expressing NSm in trans. Thus, our findings demonstrate that the NSm protein is involved in cell-to-cell spread specifically in the midgut and it is therefore a key viral determinant for successful infection of bunyaviruses transmitted via a blood meal. We also demonstrated that NSm is essential for bunyaviruses to escape from the midgut and disseminate to other tissues. How arboviruses escape from the midgut and bypass the midgut basal lamina (BL), whose pore sizes are significantly smaller than an arbovirus particle, is a long-standing question. Some evidence suggests that tracheoles penetrating the midgut BL may be actively infected by virions and provide a path for viruses to escape the midgut[32]. Other studies support the idea that passive viral escape may be facilitated due to extensive distension of the gut and transient BL "leakiness" after a blood meal[27,31,33]. Initially, only a few midgut cells become infected, but with time, viruses spread to form foci, which may eventually overlap with sporadic BL micro perforations, allowing viruses to bypass the BL and further disseminate. Therefore, the absence of BUN-ΔNSm dissemination could be due to a role of NSm in viral spreading from a midgut infected cell to a tracheole cell, similar to its function in cell-to-cell spread within the midgut epithelium. Alternatively, it could be an indirect consequence of the inability of the virus to form infection foci, decreasing the total number of infected cells and thus the chance for an infected cell to overlap with relatively few regions where the BL is compromised.

Previous reports have shown that increasing the viral dose in an infectious blood meal could overcome the midgut barriers[34,35]. We found that higher BUN-ΔNSm infectious titres in the blood meal increased the prevalence of infection as well as midgut viral titres. This is probably due to the greater number of individual cells initially infected in the midgut, and in accordance with a recent study showing that midgut viral titres shortly after invasion are strongly dose-dependent[36]. However, increased doses of BUN-ΔNSm were not associated with viral spread to neighbouring cells or with dissemination from the midgut. Thus, the midgut cell-to-cell spread and escape barriers encountered by BUN-ΔNSm are likely dose-independent. Dose-independent barriers are in general due to incompatibilities between the virus and tissue surface structures, preventing the virions from binding to and/or traversing tissues[25]. Therefore, the NSm protein may be required to overcome a structural incompatibility between the virus and midgut cell surface, allowing the virus to spread within the midgut epithelium and disseminate.

While no protein or mechanism involved in cell-to-cell spread of arboviruses in their mosquito hosts have been described, different strategies used by viruses to spread within an epithelium and egress from epithelial barriers have been characterised in other models[37,38]. In the thrip-transmitted plant bunyavirus TSWV (*Tospoviridae*), NSm acts as a plant movement protein, allowing cell-to-cell spread through junctions and long-distance movement of viral ribonucleoproteins (RNPs)[14,15,39]. The expression of NSm in this case is early and transient in intercellular junctions of TSWV-infected tissues, coinciding with the development of systemic infection, while it drastically decreases during later infection stages[40]. TSWV NSm has been shown to localise in finger-like extensions of cultured insect and plant cells[14,39]. In the midgut of BUNV-wt infected mosquitoes, we found that the NSm

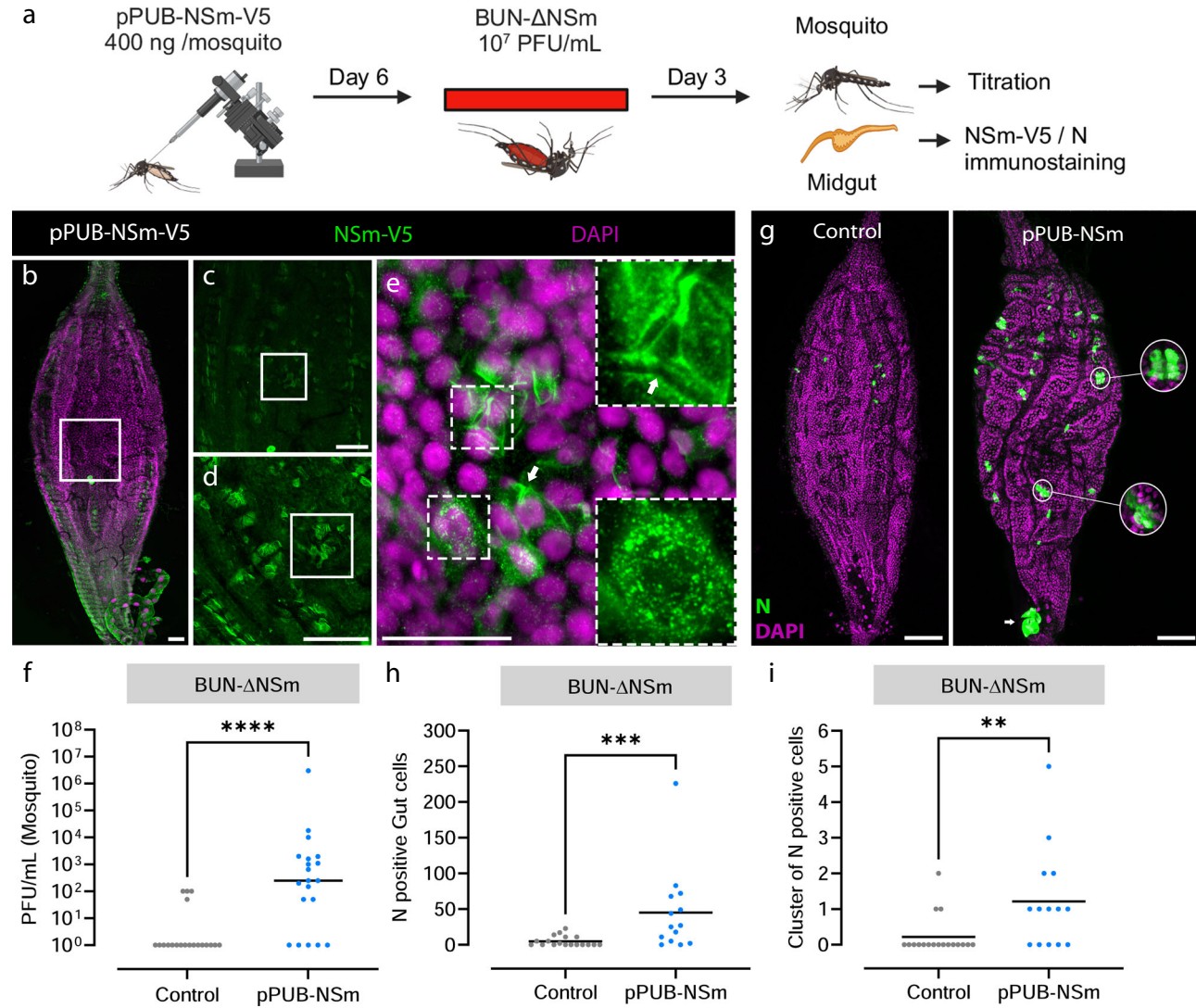

**Fig. 5 | In trans expression of NSm rescues BUN-ΔNSm infection in *Ae. aegypti*.**
**a** Workflow diagram of in vivo transfection and infection (Created in BioRender. Terhzaz, S. (2024) https://BioRender.com/f07u252). Mosquitoes were transfected with the plasmid pPUb-NSm-V5, or not transfected (controls), and fed at 6 dpi with a blood meal containing 7 x 10⁷ PFU/mL of BUN-ΔNSm, and then assessed by immunostaining and titration. Midguts were dissected at 6 dpi and stained with anti-V5 to detect the V5-tagged NSm protein (green), and with DAPI to visualize nuclei (magenta). Images show a representative midgut and are maximum projection of confocal z-series taken at (**b**) 5x, (**c**) 10x, (**d**) 20x and (**e**) 40x magnifications. Panel (**e**) contains additional dashed boxed areas to illustrate the cytoplasmic NSm punctate staining as well as the plasma membrane of midgut epithelial cells (arrow). Scale bars are 50 μm. **f** Virus titres of whole mosquitoes at 3 dpbm were determined by plaque assay on BHK-21 cells and displayed as PFU/mL with all

individual animals that were infected displayed ($n = 20$ per condition). Lines indicate median values. Statistical testing by two-tailed Mann-Whitney test for viral titres. ****$p$ value < 0.0001. Statistical analysis of the infection prevalence for controls (20 %) and pPUb-NSm-V5 (75 %) was performed using a Chisquare test. ***$p = 0.0006$. **g–i** Mosquitoes were microinjected with the recombinant plasmid pPUb-NSm, or not (controls), and fed at 6 dpi with a blood meal containing 2 x 10⁸ PFU/mL of BUN-ΔNSm. **g** Merged z-stack representative images of midguts stained at 6 dpbm with an N antibody (green) and DAPI (magenta). Scale bars are 150 μm. Quantification of individual (**h**) or cluster of (≥3 cells/foci, (**i**) N positive cells per midgut of infected female mosquitoes ($n = 18$ and 14 for controls and pPUb-NSm respectively) at 3 dpbm. Lines indicate mean values. Statistical testing by two-tailed Mann-Whitney test. **$p$ value = 0.006 and ***$p$ value = 0.001. Source data are provided as a Source Data file.

protein was expressed mainly at the periphery of infection foci. These data suggest an early and transient expression of NSm in infected cells and it is consistent with the role of NSm in virus spread within the midgut epithelium. In addition, NSm was localised at the cell membrane in the midgut, where cell junctions are located, and localised in finger-like extensions in mosquito cultured cells. The similar characteristics between TSWV NSm and BUNV NSm suggest that the latter facilitates virus cell-to-cell spread by a comparable mechanism to that of TSWV NSm.

Interestingly, infection with Rift Valley fever virus (RVFV) mutated for P78, a protein in which NSm remains fused to Gn, was also associated with reduced infection rates and dissemination in *Ae. aegypti*

mosquitoes[41,42]. Although no mechanism was identified, the mutant virus was also restricted to small areas in the midgut[43]. Thus, the critical function of NSm in cell-to-cell spread may be a conserved feature of different bunyavirus families. Our analysis showed that NSm is present in bunyaviruses belonging to only a few viral genera and which are spread across phylogenetically distinct families supporting that NSm acquisition is the result of convergent evolution as these viruses were adapting to similar environments and selective pressures. Since NSm is present in arbo-bunyaviruses only, one of the possible explanations is that these viruses have acquired NSm to overcome the insect vector midgut barrier after oral infection. Remarkably, NSm is absent in insect-specific bunyaviruses. While arboviruses are acquired through a

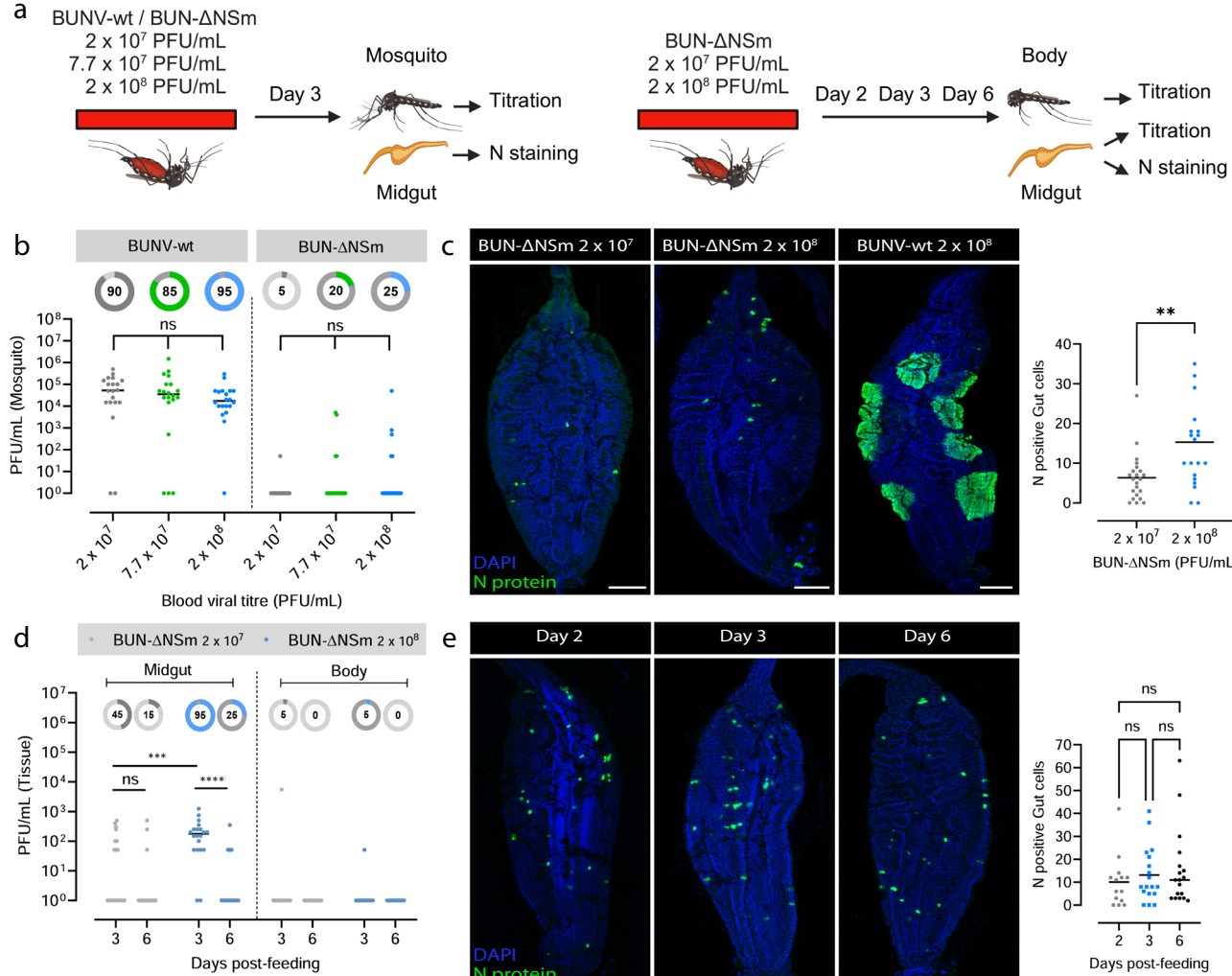

**Fig. 6 | High doses of BUN-ΔNSm viruses do not impact dissemination in *Ae. aegypti*. a** Workflow diagram (Created in BioRender. Terhzaz, S. (2024) https://BioRender.com/s43c481). **b** Mosquitoes were fed with $2 \times 10^7$, $7 \times 10^7$ or $2 \times 10^8$ PFU/mL of either BUNV-wt and BUN-ΔNSm, and virus titres determined. Titres are displayed as PFU/mL for each whole individual female ($n = 20$ per condition). Statistical significance shown on the graph was obtained using a two-way ANOVA followed by a Tukey's multiple comparisons test. ns, not significant, $p$ value > 0.5. Lines indicate median values and the circled numbers represent the percentage of infected samples. **c** Merged z-stack confocal images of midguts infected by $2 \times 10^7$ or $2 \times 10^8$ PFU/mL of BUN-ΔNSm and $2 \times 10^8$ PFU/mL of BUNV-wt and stained with an N antibody (green) and DAPI (blue). Scale bars are 150 μm. Total number of N positive cells per midgut of infected female mosquitoes with $2 \times 10^7$ ($n = 22$) or $2 \times 10^8$ PFU/mL ($n = 20$) of BUN-ΔNSm. Statistical testing by two-tailed Mann-Whitney test. **$p = 0.04$. Lines indicate mean values. **d** Mosquitoes were fed with either $2 \times 10^7$ or $2 \times 10^8$ PFU/mL of BUN-ΔNSm, and individual midguts and rest of bodies virus titres at 3 and 6 dpbm were determined ($n = 20$ per condition). Statistical testing by a two-way ANOVA followed by a Tukey's multiple comparisons test. ns, not significant, $p$ value > 0.5; ***$p$ value = 0.0002, ****$p < 0.0001$. Lines indicate median values and the circle number represent the percentage of infected samples. **e** Confocal images of midguts infected by $2 \times 10^8$ PFU/mL of BUN-ΔNSm and stained at 2, 3 and 6 dpbm with an N antibody (green) and DAPI (blue). Scale bars are 150 μm. Total number of N positive cells per midgut of infected female mosquitoes at 2 ($n = 14$), 3 ($n = 18$) and 6 ($n = 18$) dpbm. Statistical testing by Kruskal-Wallis. ns, not significant, $p$ value > 0.5. Lines indicate mean values. Source data are provided as a Source Data file.

blood meal and need to move efficiently between cells in the midgut and then disseminate to other parts of the mosquito body, insect-specific viruses (ISVs) are differently acquired, such as by vertical or venereal transmission[44]. Therefore, ISVs could have evolved different mechanisms suited to their acquisition modes where direct access to mosquito tissues, and their maintenance in mosquitoes, bypasses the midgut challenges faced by arboviruses, making the NSm protein unnecessary.

Although there is a strong correlation between the presence of an NSm protein and arbo-bunyaviruses transmitted by insects, this correlation does not strictly extend to tick-borne bunyaviruses, *e.g.*, *Bandavirus* and *Banyangvirus* genera. One plausible reason may be that these tick-borne viruses do not have to overcome similar midgut barriers in their tick hosts. The lack of comprehensive

knowledge about how bunyaviruses infect and disseminate in ticks in vivo complicates the formulation of data-driven hypotheses to explain why insect-borne bunyaviruses evolved to possess NSm while some tick-borne bunyaviruses did not. However, several speculations can be proposed. For instance, tick-borne bunyaviruses may not require cell-to-cell spread in the tick gut for propagation and dissemination. Interestingly, while hematophagous insects digest blood in their gut lumen, in ticks, blood digestion occurs intracellularly. This intracellular absorption of blood - and thus virus particles - into gut cells could lead to a higher number of cells being initially infected, resulting in a productive gut infection without the need for viral cell-to-cell spread. Additionally, biological differences such as the prolonged tick blood-feeding process and significant, long-lasting gut distension may favour passive viral

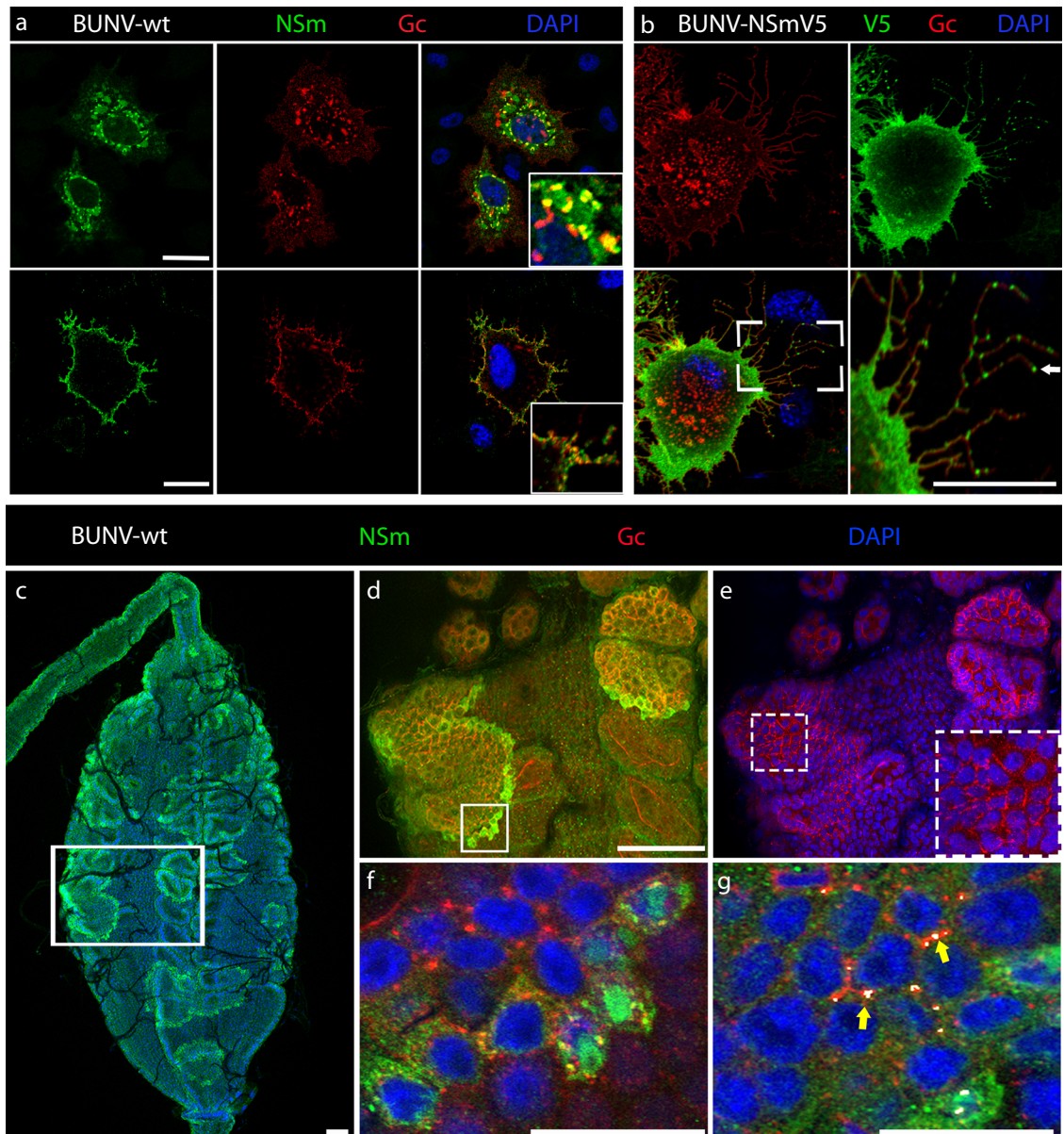

**Fig. 7 | NSm is expressed on the cell surface of infected cells and at the periphery of foci in BUNV infected *Ae. aegypti* midguts. a** Internal and surface expression of NSm and Gc in BUNV-wt infected (MOI 0.01) C6/36 cells permeabilized (for internal staining, top images) or not permeabilized (for surface staining, bottom images). Cells stained with anti-Gc (red), anti-NSm (green), counterstained with DAPI (blue) show partial co-localisation (yellow). Maximum projection of confocal z-series with inserts showing higher magnification images. **b** Cell surface staining of C6/36 cells infected with recombinant BUNV-NSmV5 virus. Cells were stained with anti-Gc (red), anti-V5 (green), and counterstained with DAPI (blue). Scale bar, 10 μm. **c–g** Midguts dissected from females at 3 dpbm (BUNV-wt titre in the bloodmeal: 7 × 10⁷ PFU/mL) were co-stained with BUNV-specific NSm (green) and Gc (red) antibodies and with DAPI to visualize nuclei (blue). **c** Maximal intensity z-projection of a whole midgut shows intense NSm immunoreactivity at the periphery of each infection foci. High-magnification image of the boxed area in (**c**) showing merged z-projections images of (**d**) NSm and Gc immunoreactive signals and (**e**) Gc and DAPI signals. Panel (**e**) contained an additional dashed boxed area to illustrate the Gc staining distributed around the circumference of cells inside the foci of infection. **f** Individual slice image of the boxed area in (**d**) shows partial co-localisation between NSm and Gc in infected cells at the periphery of the focus of infection. **g** Merged z-projection image analysed using Imaris to highlight 3D co-localisation regions between NSm and Gc immunoreactive signals. Both NSm and Gc are co-localised at the junction between infected cells as shown in white by the co-localisation model (yellow arrow). All scale bars are 100 μm.

escape - prolonged BL "leakiness" - and explain why tick-borne viruses can disseminate without NSm.

Since the role of NSm in cell-to-cell spread seems to be conserved across bunyavirus families, it would be interesting in a future study to test for complementation of BUN-ΔNSm cell-to-cell spread using heterologous NSm proteins. Our protein sequence homology and phylogenetic analyses revealed that the presence of NSm in arbo-bunyaviruses is likely the result of independent ancestral acquisition of different genes, all presently called NSm, by the

different virus families. Therefore, trans expression may work between viruses of the *Peribunyaviridae* family but may not work between bunyavirus families. Since different proteins can be involved in a same cell-to-cell spread mechanism and different cell-to-cell spread mechanisms exist[37], a lack of rescue would not exclude that those different independently acquired genes still exert a similar function in cell-to-cell spread. Additional studies will be required to validate a conserved role of NSm in cell-to-cell spread after oral transmission, to characterise the exact cell-to-cell spread

mechanism, and to understand why some tick-borne bunyaviruses do not encode NSm.

Our findings fill a gap in our understanding of how some bunyaviruses evolved to be transmitted by blood sucking insects. We demonstrate that NSm is required for virus cell-to-cell spread in the midgut and to escape the midgut barrier, thereby being necessary for the establishment of productive infection and dissemination in the mosquito host after a blood meal. This study also has important translational applications. Since live-attenuated vaccine strains against arboviruses should not spillover in the vectors (*e.g.*, one of the WHO requirements for RVF vaccines[45]), the elucidation of the function of NSm as a crucial viral determinant for mosquito infection, dissemination and transmission could reinforce the use of viruses lacking NSm for the development of live-attenuated vaccines to prevent infections caused by mosquito-borne bunyaviruses.

## Methods

### Cell culture

*Ae. albopictus* C6/36 and U4.4 cell lines (derived from larvae[46,47], obtained from Prof. A. Kohl, LSTM, UK), and *Ae. aegypti* Aag2 cell line (derived from embryonic tissue[46], obtained from Prof. P. Eggleston, Keele University, UK) were maintained in Thermo Fisher Scientific tissue culture 25 cm² flasks with Leibovitz's L-15 media (Gibco) supplemented with 10% (w/v) Tryptose Phosphate Broth (TPB; Gibco) and 10% (w/v) foetal bovine serum (FBS; Gibco). BHK-21 cells (obtained from Prof. R. M. Elliott, MRC-University of Glasgow Centre for Virus Research, UK) and BSR-T7/5 cells[48] were cultured in Glasgow minimal essential medium (GMEM, Gibco), supplemented with 10% FBS, 10% TPB (Gibco) and 83 U/mL penicillin/streptomycin (Gibco). A549 and A549/NPro cell lines (obtained from Prof. R. E. Randall, University of St. Andrews, UK) were maintained in Dulbecco's modified Eagle's medium (DMEM, Gibco), supplemented with 10% FBS and 2 µg/mL puromycin (Gibco). A549, A549/Npro, BSR-T7/5 and BHK-21 cells were grown at 37 °C with 5% CO2, and C6/36, U4.4 and AF5 cells were cultured at 28 °C.

### Plasmids, generation of recombinant viruses by reverse genetics and virus growth curves

Rescue experiments were performed as described previously[49]. Briefly, BSR-T7/5 cells were transfected with a mixture of plasmids that generate full-length BUNV wild type antigenome RNA transcripts (pT7riboBUNL, pT7riboBUNM, and pT7riboBUNS)[49,50], or pTM-BUNM ΔNSm_I which contain deletion of the coding region for mature NSm and Gc signal sequence (residues 332 to 477) of BUNV GPC for rBUN-ΔNSm (Fig. S2a[17],). Stocks of BUNV-wt, BUN-ΔNSm, and BUNV-NSmV5 (V5 epitope inserted between residues 403 and 420 of the NSm coding region)[17] were grown and titrated in BHK-21 cells as previously described[51]. To monitor virus growth, BSR-T7/5 cells in 6-well plate were infected at a multiplicity of infection (MOI) of 0.01 PFU per cell. Supernatants were harvested at various time points as indicated and stored at −80 °C until virus titration. To generate the pPUb-NSm-V5 and pPUb-NSm plasmids, the full NSm ORF was amplified from the pT7riboBUNM plasmid with KOD Hot Start Master Mix (EMD Millipore) using the primers listed in Table S1, allowing the insertion of a start codon before NSm domain I and the V5 sequence (or not) downstream of NSm domain V. These fragments were further cloned with In-Fusion cloning (Takara Biotech) downstream of the polyubiquitin promoter into the pPub plasmid[52,53]. Final constructs were verified via Sanger sequencing.

### In vivo transfection of plasmids expressing BUNV NSm gene

To express NSm in the mosquito midguts, we used a previously published method[54], with slight modifications. Briefly, 400 ng of pPUb-NSm-V5, pPub-NSm, or pPub-Gal4 (negative control for V5 staining) per female mosquito were gently mixed with Cellfectin II (Thermo

Fisher Scientific) as 1:1 ratio (vol/vol) and incubated for 20 min at room temperature prior to thoracic microinjection of 1- and 2-day-old cold-anaesthetised female mosquitoes using a nanoinjector (Nanoject III, Drummond Scientific). Mosquitoes were allowed to recover at 28 °C and 80% humidity with access to 10% (w/vol) sucrose solution *ad libitum* until BUNV ΔNSm infectious blood meal.

### Mosquito rearing

*Aedes aegypti* Paea strain (a gift of Dr A-B. Failloux, Institut Pasteur, France) was reared at 28 °C and 80% humidity conditions with a 12 h light/dark cycle. *Ae. aegypti* eggs on filter paper were placed in trays containing ~1.5 cm of water to hatch overnight. Larvae were fed with dry cat food (Friskies) until pupation. Emerging adult mosquitoes were transferred into BugDorm mosquito cages and maintained on a 10% (w/vol) sucrose solution *ad libitum*. Females were fed with heparinised rabbit blood (Envigo, UK) for 1 h using a 37 °C Hemotek system (Hemotek Ltd).

### Virus production for mosquito infections

To amplify virus stocks, BHK-21 cells in T225 flasks were infected with P1 viruses at a MOI of 0.015 PFU per cell and incubated for 3 days at 33 °C in GMEM supplemented with 2% FBS and 10% TPB. Upon harvest, the supernatant was clarified by centrifugation at 4500 *g* for 10 min and filtered using a 0.22 µm Stericup vacuum filter unit (EMD Millipore). The supernatant was then concentrated by ultracentrifugation at 140000 *g* for 2 h at 4 °C. The pellet was resuspended in GMEM medium supplemented with 10% FBS and 2.5% Sodium Bicarbonate (Gibco). The virus stock was aliquoted and stored at −80 °C. For titration, 10-fold serial dilutions of virus stock were prepared in Opti-MEM containing 2% FBS. Following an hour-long incubation with inoculum, the cells were overlaid with 1X MEM (Gibco) containing 2% FBS and 0.6% Avicel (FMC Biopolymer). The cells were incubated for 3 days at 37 °C with 5% CO2, and thereafter fixed using equal volume of 8% formaldehyde solution for 1 h. Plaques were stained using 0.1% toluidine blue (Sigma-Aldrich). The titre of infectious particles is expressed as plaque forming units per mL (PFU/mL).

### BUNV mosquito infections

For oral infection assays, 5-day-old female mosquitoes were starved 24 hours prior to being offered an infectious blood-meal. Frozen BUNV stocks were thawed at 37 °C and diluted with PBS-washed fresh rabbit blood (Envigo) to give a final titre of 7.7 x 10⁷ PFU/mL unless otherwise stated and supplemented with 10 mM final ATP. Females were allowed to feed for at least 30 min, after which mosquitoes were cold anaesthetized, and only engorged females were further kept at 28 °C and 80% humidity with access to 10% (w/vol) sucrose solution *ad libitum* until sampling. Some engorged females (*n* = 5 individual females per condition) were sampled just after the infectious blood meal, homogenized in 100 µL of L-15 containing 10% FBS and stored at −80 °C until titration to assess ingested virus quantity.

For infection assays by inoculation, individual mosquitoes aged 5 to 7 days post-emergence were injected intrathoracically with 5 x 10⁴ PFU/mosquito (corresponding to BUNV-wt titres measured in bodies following an oral infection) using a Nanoject II Auto-Nanoliter injector (Drummond Scientific). Mosquitoes were allowed to recover at 28 °C and 80% humidity with access to 10% (w/vol) sucrose solution *ad libitum* until sample collection.

### Salivation and perfusion of hemocytes

For salivation experiments, individual mosquitoes at 7 days post-infection were used. Mosquitoes were cold anaesthetized, and legs and wings removed. At room temperature, the mosquito proboscis was placed in a 10 µL filter tip containing 5 µL of serum. Mosquitoes were then left to salivate for 30 min. The serum containing saliva was then expelled into 45 µL of L-15 media, placed on dry ice and stored at

−80 °C until titration. For perfusion experiments, circulating cells from 7-day post-infection females were collected by perfusion. Briefly, mosquitoes were cold-anaesthetized, and the last abdominal segment was removed. With a microinjection needle, 1× PBS was injected in the thorax, and the first five drops exiting from the abdomen were collected on an ibidi treated μ-slide 8 well chamber (5 perfused mosquitoes per well). After 30 min of cell attachment, cells were fixed and stained as described below.

## Virus titration in mosquito samples

Tissues dissected in RNase free 0.05% PBS-Tween 20 (PBST) (v/v) or whole females were transferred into tubes containing sterile glass microbeads (0.5 mm; Sigma-Aldrich) and 100 μL of Leibovitz's L-15 Medium supplemented with 10% FBS. Samples were homogenized using a Precellys 24 (Bertin Technologies) at 6500 bpm for 30 sec and stored at −80 °C until titration. Mosquito samples and saliva samples were thawed at 37 °C and spun for 2 min at 10,000 x $g$. Supernatant was serially diluted 10-fold in OptiMEM containing 2% FBS and 1% antibiotic-antimycotic solution (100 U/mL penicillin, 0.1 g/mL streptomycin, 250 ng/mL amphotericin B, Sigma-Aldrich) and inoculated onto BHK-21 cells seeded the day before at $1.8 \times 10^5$ cells/mL on 12-well plates. Plates were incubated for 1 h at 37 °C with 5% CO2 and the cells were overlaid with 1X MEM (Gibco) containing 2% FBS, 1% antibiotic-antimycotic and 0.6% Avicel (FMC Biopolymer). The cells were incubated for 3 days at 37 °C with 5% CO2, and thereafter fixed using equal volume of 8% formaldehyde solution for 1 h. Plaques were stained using 0.1% toluidine blue (Sigma-Aldrich). The titre of infectious particles is expressed as a concentration in PFU/mL for each sample. Since samples could not be measured without dilution, detection limit is 50 PFU/mL (i.e., $10^{1.69}$ PFU/mL). To present titres including negative ones in log scale, +1 was added to all values including the negative ones (i.e., $1 = 10^0$). Undetectable samples (no plaque at the first dilution, below the detection limit $10^{1.69}$ PFU/mL) are plotted at $10^0$ PFU/mL. Absolute quantity in each sample is 10 times less for all samples (sampled in 0.1 mL) and 20 time less for saliva samples (sampled in 0.05 mL).

## RNA extraction and reverse transcription - quantitative PCR (RT-qPCR)

Tissues were dissected from 5-day-old females in RNase free PBS with 0.05% Tween 20 and homogenized in 1 mL of TRIzol (Thermo Fisher Scientific) and stored at −80 °C ($n = 20$ per condition and per independent experiment). RNA extraction was performed according to the manufacturer's protocol except that 1 M 1-Bromo-3-ChloroPropane (BCP) (Sigma-Aldrich) was used instead of chloroform. DNase treatment was performed for 30 min at 37 °C following the manufacturer's protocol (TURBO DNase-free kit, Thermo Fisher Scientific), except that RNasin 0.36 U/μL (Promega) was also added. Complementary DNA was synthesized from 500 ng of total RNA in a 20 μL final volume using the High-Capacity cDNA Reverse transcription kit (Applied Biosystems). qPCR was performed with the QS3 Real Time PCR System (Applied Biosystems) using the Fast SYBR Green Master Mix method (Thermo Fisher Scientific) according to the manufacturer's protocol and using specific primers (Sigma-Aldrich) listed in Table S1. To quantify BUNV S RNA levels in midgut samples from BUNV-wt or BUN-ΔNSm fed mosquitoes, qPCR assays were run with the comparative Ct (cycle threshold) method using S7 ribosomal protein gene as a standard gene for normalisation and according to the Taylor method[55] to obtain a geomean of RQ = 1 for the control group (BUNV-wt 3 h) and relative RQ values for every other sample.

## Immunocytochemistry

Immunofluorescence assays were performed as previously described[56]. In brief, mosquito tissues were dissected in PBS-Tween 0.05% and fixed in 4% (wt/vol) paraformaldehyde in PBS for 30 min at room temperature. BHK-21 and C6/36 cells seeded the day before on an ibidi treated μ-slide 8 well chamber (2 and $5 \times 10^4$ cells/well respectively) as well as perfused hemocytes were fixed for 15 min at room temperature. For surface staining, cells were washed with PBS and incubated for 1 h in blocking solution (10% normal goat serum in PBS). For internal staining, fixed tissues and cells were permeabilised with PBS-Triton 100X 0.1% and blocked for 1 h in 10% normal goat serum in PBS-Triton 100X 0.1%. Tissues were then incubated with primary antibodies in blocking buffer at 4 °C overnight, followed by overnight incubation in secondary antibodies in blocking buffer at 4 °C. Counter staining with 1 μg/mL DAPI (Sigma-Aldrich) and/or phalloidin 488/647 nm (1:100; ThermoFisher) were performed where appropriate. The primary antibodies used were mouse anti-V5 (1:500, Abcam), rabbit anti-NSm (1:100[57],), rabbit anti-N (1:500[51],), mouse anti-Gc (1:500[58],). The secondary antibodies used were goat-anti-mouse/rabbit Alexa 488/546 (1:1000; ThermoFisher). Tissues were mounted between a slide and coverslip with an imaging spacer (Sigma-Aldrich) using ibidi Mounting Medium (ibidi). Images were acquired on a Zeiss LSM 880 inverted confocal microscope with Airyscan using the Zen or Imaris image analysis software packages and processed with Adobe Photoshop.

## Phylogeny analysis

The mature peptide sequence of BUNV NSm was used to blast v2.11.0 using tblastn (PMID: 9254694) against all Bunyavirales reference genomes downloaded from Genbank on the 22nd of July 2024. The protein alignment of NSm was produced for all species with a match using MAFFT v7.475 (PMID: 23329690). Non-homologous gap regions of the alignment were removed using trimal v1.5 with option -gappyout (PMID: 19505945). The maximum likelihood tree was reconstructed in IQTREE version 2.1.3 (PMID: 32011700) using the best model based on ModelFinder and ultrafast bootstrap with 1000 replicates. The phylogeny and the alignment were visualized in R v4.4.0 using the packages APE (PMID: 30016406) and ggtree (PMID: 32162851) and pairwise distance calculated with the seqinr package[59]. Structures were predicted for the NSm sequences of BUNV, RVFV, CCHFV and TSWV using the ColabFold (v1.5.5) implementation of AlphaFold2 (v2.3)[60].

## Statistics and reproducibility

Statistical analysis was performed using a statistical software package (GraphPad Prism 9). For number of cells and virus titres, the non-parametric Mann-Whitney or Kruskal-Wallis tests were used to evaluate the significance of the results between two or three groups (with Dunn's multiple comparison test). For experiments with two variables, two-way ANOVA followed by Tukey's multiple comparison test was used. For virus titres, statistics were performed after $\log_{10}$ transformation of the data. Statistical analyses of the infection prevalence were performed using a Chisquare test using the proportion of infected samples and the number of total samples per analysed group. For relative mRNA detection using the comparative Ct method, Log2-transformed values of RQ values were used for two-way ANOVA analysis followed by Tukey's multiple comparisons. All differences were considered significant at $p$ value < 0.05. All plots have statistical significance indicated as follows: *$p$ value < 0.05, **$p$ value < 0.01, ***$p$ value < 0.001, ****$p$ value < 0.0001, and ns = not significant. All experiments have been repeated independently with similar results at least two times.

## Reporting summary

Further information on research design is available in the Nature Portfolio Reporting Summary linked to this article.

# Data availability

Source data are provided with this paper and are available in the University of Glasgow repository (https://doi.org/10.5525/gla.researchdata.1516). Source data are provided with this paper.

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

## Acknowledgements

This study was funded by the UK Medical Research Council (MC_UU_12014/8 and MC_UU_00034/4 to EP and AK; MC_UU_00034/8 to EP; MC_UU_00034/7 to CVR Imaging Platform; https://mrc.ukri.org/).

We acknowledge Melanie McFarlane (MRC-University of Glasgow Centre for Virus Research) for critical reading of this manuscript. We thank Yubing Chen and Anna Krumbein for technical assistance. We are grateful to all members of Pondeville, Kohl and Brennan groups for helpful discussions. Workflow figures were created with Biorender.com (full licence).

## Author contributions

S.T.: Conceptualization, Data curation, Formal Analysis, Investigation, Methodology, Validation, Visualization, Writing – original draft, Writing – review & editing. D.K.: Investigation, Project administration, Writing – review & editing. F.A.: Investigation, Writing – review & editing. A.M.S.: Investigation, Writing – review & editing. J.H.: Investigation, Writing – review & editing. J-P.P.: Investigation, Writing – review & editing. M.P.: Supervision, Writing – review & editing. A.K.: Funding acquisition, Writing – review & editing. X.S.: Conceptualization, Data curation, Formal Analysis, Investigation, Methodology, Visualization, Writing – original draft, Writing – review & editing. E.P.: Conceptualization, Data curation, Formal Analysis, Investigation, Funding acquisition, Methodology, Project administration, Supervision, Validation, Visualization, Writing – original draft, Writing – review & editing.

## Competing interests

The authors declare no competing interests.
