## [Transparent Peer Review file · Nature Communications]

NSm is a critical determinant for bunyavirus transmission between vertebrate and mosquito hosts

Corresponding Author: Dr Emilie Pondeville

Version 0:

Reviewer comments:

Reviewer #1

(Remarks to the Author)

This is a fantastic manuscript addressing one of the major unanswered questions in the field of bunyavirology – namely, the function of the NSm protein that a subset of these viruses encode. This manuscript addresses the function of NSm in *Aedes aegypti* mosquitoes. The authors thoroughly examined the role of NSm in vitro and in vivo and identified that NSm deleted BUNV is unable to efficiently spread outside of the midgut and to the salivary glands, thus transmission from mosquito to mammal would not be able to occur. The stunning immunofluorescence staining in the midgut shows that NSm is critical for efficient cell-to-cell spread in the midgut. This is an important finding for the field with translational relevance given the use of NSm deleted strains for bunyavirus vaccination. Mostly minor changes are recommended, apart from one major comment concerning the conclusions of the in trans complementation of NSm. The paper is super well written and a joy to read, which is refreshing. The imaging is fantastic and the coup de grace is the trans addition of NSm and partial restoration of infection. My only major comment is about the title, which I feel does not represent the findings of the study. Perhaps “NSm is a critical determinant for bunyavirus dissemination in mosquitoes.”

Major comments:

Clarification is needed regarding the control for Figure 5 in text, the methods, and figure legend. Were the control mosquitoes in Fig 5G also microinjected with a control plasmid and the transfection reagent, Cellfectin II? If not, please explain whether Cellfectin II or the microinjection process could alter the conclusions from this experiment and the overall study. I.e. Could Cellfectin exposure reduce the barriers within the midgut, thus explaining the recovered phenotype?

Minor comments:

1. The Fig 1 finding that primarily arthropod-transmitted bunyaviruses encode NSm is intriguing. It would be good for the authors to speculate in the discussion why Bandavirus and Banyangvirus do not appear to encode an NSm protein.
2. Also related to Fig 1: Are there any sequence similarities between the NSm proteins encoded by the divergent bunyaviruses? It would be helpful for the authors to comment on this when discussing Fig. 1 or even provide an alignment in the supplementary figures. It is outside the scope of this manuscript, but it would be interesting to see if trans expression of NSm from a different bunyavirus could also rescue BUN.
3. Can the authors clarify the bloodmeal doses chosen for these studies. Fig. 2 uses 4×10^8 pfu, Fig 3 uses 5×10^4 , etc. How were the doses selected for each route/experiment?
4. The doses used in Fig 6 to address overcoming the midgut barrier are only 10-fold different and thus make it hard to definitively conclude that dose has not impact on dissemination.
5. Should line 204 read to lower levels for DelNSm compared to wt (phrase reversed)? Line 119: It would be helpful to describe in text (outside of methods) and in figure/legend which species and organ these cell lines
6. Figure 4 legend: define the LOD for qRT-PCR. Also define in methods
7. Figure S2: please include the statistical method used to determine no significant difference in S2A/B.
8. Multiple minor typos and discrepancies in how acronyms or measurements are written in text throughout (i.e. ml vs mL).

Reviewer #2

(Remarks to the Author)

I co-reviewed this manuscript with one of the reviewers who provided the listed reports. This is part of the Nature Communications initiative to facilitate training in peer review and to provide appropriate recognition for Early Career

Researchers who co-review manuscripts.

Reviewer #3

(Remarks to the Author)

The manuscript titled "NSm is a critical determinant for bunyavirus transmission between vertebrate and arthropod hosts" by Terhzaz S. et al. describes that the Bunyamwera virus (BUNV) nonstructural protein NSm is dispensable for viral replication in mosquito cell culture, but necessary for viral replication and spread in the midgut cells of *Aedes aegypti* mosquitoes. Meanwhile, intrathoracic injection of BUNV lacking the NSm gene resulted in efficient viral spread, indicating a specific function of NSm in the midgut cells. The authors also showed that (i) the in trans expression of NSm facilitates the replication of BUNV lacking the NSm gene, and (ii) increasing the dose of BUNV lacking NSm does not lead to viral spread in the midgut. Imaging of NSm protein in the mosquito midgut epithelium was also performed. The manuscript is generally well-written, and this study will provide new insight into the function of the orthobunyavirus NSm protein in mosquito vectors. However, apparently, the interpretation of the experimental result beyond the orthobunyaviruses has not been validated. For example, in the case of Rift Valley fever virus (RVFV), it is the 78kD protein, not the NSm protein, that has been shown to be responsible for dissemination via the mosquito midgut. In terms of technical aspects, it is important to include the threshold of detection for plaque assays and to ensure the inclusion of mock-infected controls for the validation of immunological detection of viral antigens. Specific points are as follows:

Major points

- Lines 101 – 110 and Figure 1: The authors generalized the presence of NSm proteins across arboviruses, including the Phenuiviridae. However, for instance, RVFV encodes a 78kD protein for viral dissemination via the midgut of *Aedes aegypti* mosquitoes, whereas specific deletion of NSm did not impair viral dissemination via the midgut (Kreher et al., *Emerg. Microbes Infect.* 2014, 3: e71). Meanwhile, the presence of 78kD-like proteins in other phleboviruses is poorly characterized. Given this complexity, direct comparison of phenuiviridae NSm with that of BUNV or other bunyaviruses in this study seems challenging.
- Similar to the comment above, insect-specific bunyaviruses like Gouleako virus do not encode the NSm protein. Further elaboration on the potential distinctions between arboviruses and insect-specific bunyaviruses regarding virus transmission would strengthen the rationale of this study.
- Figure 3e, f, g: negative controls (mock-infected) should be included to validate the detection of the virus N antigen.
- Figure 2b-c, 3b-d, 4b, 5f, 6b, d, and S2a-b: In the plaque assay, is it possible to detect 1 PFU/ml (i.e., 10^0 PFU/ml)? If not, please specify the limit of detection in each graph.
- Figure 5. Mock-transfected cells should be stained with anti-V5 to validate the immunostaining assay using midguts. In addition, authors should describe how intrathoracic injection of transfection mixture can reach to midgut epithelium: e.g., crossing basal lamina?

Minor points

- Lines 89-90, 101 – 110: "arbovirus" should be defined as plant virus is also included in Figure 1.
- Please define the "PFU/ml (Mosquito)" in graphs.
- Line 200: "Interestingly, in contrast to virus titres, the levels of viral segment S levels were similar for both viruses at 3 and 24h, which suggests that viral RNA from initially ingested virions persists in the midgut for 24h post-blood meal, likely in the gut lumen while the blood is not yet fully digested." This does not fully account for the absence of infectious virus at 24 hpbm.
- Line 297: "BUNV-NSmV5" should be explained further in terms of the location of V5-tag insertion.
- Line 404-405: Please specify "RVFV lacking both 78kD and NSm genes".
- Line 407: "Thus, the critical function of NSm in cell-to-cell spread may be a conserved feature of different bunyavirus families." By ignoring the 78kD function, this description is not very logical.
- Figure S1a: The illustration of BUN M requires further clarification in terms of the deletion site of NSm region: e.g., $\Delta 303 - 331$?

Version 1:

Reviewer comments:

Reviewer #1

(Remarks to the Author)

No comments, all of my concerns were addressed.

Reviewer #2

(Remarks to the Author)

I co-reviewed this manuscript with one of the reviewers who provided the listed reports. This is part of the Nature Communications initiative to facilitate training in peer review and to provide appropriate recognition for Early Career

Researchers who co-review manuscripts.

Reviewer #3

(Remarks to the Author)

The revised manuscript titled "NSm is a critical determinant for bunyavirus transmission between vertebrate and arthropod hosts" by Terhzaz S. et al. has addressed most of the reviewers' concerns. However, two minor issues remain that should be addressed to maintain scientific accuracy.

Specific points:

1. Figure 1 legend: The statement "Phleboviruses (e.g., Rift Valley Fever virus) also encode P78, which consists of an NSm-Gn fusion protein" is not scientifically accurate. The 78kD protein is not identical to the NSm-Gn fusion protein, as it can also be expressed separately from NSm or the 78kD protein. The 78kD protein contains a signal peptide and is likely structurally distinct from the NSm protein or the NSm-Gn protein.
2. Figures 2b-c, 3d, 4b, 5f-i, 6b, d, and S3a-b: The Y-axis should include a threshold line, and undetected samples should be plotted as "< 50 PFU/ml" or "undetectable." This reviewer believe that the plotting these onto 10^0 is not scientifically accurate.

REVIEWER COMMENTS

We sincerely thank the reviewers for their valuable and constructive feedback on our manuscript. We have carefully addressed each comment and provided a detailed, point-by-point response (R) below. The manuscript has been revised accordingly.

Reviewer #1 (Remarks to the Author):

This is a fantastic manuscript addressing one of the major unanswered questions in the field of bunyavirology – namely, the function of the NSm protein that a subset of these viruses encode. This manuscript addresses the function of NSm in *Aedes aegypti* mosquitoes. The authors thoroughly examined the role of NSm in vitro and in vivo and identified that NSm deleted BUNV is unable to efficiently spread outside of the midgut and to the salivary glands, thus transmission from mosquito to mammal would not be able to occur. The stunning immunofluorescence staining in the midgut shows that NSm is critical for efficient cell-to-cell spread in the midgut. This is an important finding for the field with translational relevance given the use of NSm deleted strains for bunyavirus vaccination. Mostly minor changes are recommended, apart from one major comment concerning the conclusions of the in trans complementation of NSm. The paper is super well written and a joy to read, which is refreshing. The imaging is fantastic and the coup de grace is the trans addition of NSm and partial restoration of infection.

My only major comment is about the title, which I feel does not represent the findings of the study. Perhaps “NSm is a critical determinant for bunyavirus dissemination in mosquitoes.”

(R): We thank the Reviewer for their positive comments on our manuscript. Regarding the title, we feel that “dissemination” could be misleading as NSm is indeed required for propagation in the gut and dissemination from the gut, however not in other body parts (e.g., injected virus disseminates to the salivary glands). Since our findings show that NSm is required to initially produce a productive gut infection following an infectious blood meal – transmission from vertebrate to mosquito, and also required for ultimate dissemination – transmission from mosquito to vertebrate, we changed the title to “NSm is a critical determinant for bunyavirus transmission between vertebrate and mosquito hosts”, which we believe represents our findings better with regards to the mosquito host, as opposed to “arthropod hosts”.

Major comments:

Clarification is needed regarding the control for Figure 5 in text, the methods, and figure legend. Were the control mosquitoes in Fig 5G also microinjected with a control plasmid and the transfection reagent, Cellfectin II? If not, please explain whether Cellfectin II or the microinjection process could alter the conclusions from this

experiment and the overall study. I.e. Could Cellfectin exposure reduce the barriers within the midgut, thus explaining the recovered phenotype?

(R): We apologise for the lack of clarity in the manuscript. The control mosquitoes in Fig. 5g were not injected with Cellfectin. The legend of Fig. 5 has thus been modified accordingly. Cellfectin is a liposome-based reagent widely used for DNA transfection into insect cell lines. Liposomes are biocompatible, biodegradable, non-immunogenic, and well-known for their instability and their short half-life in biological environments. They fuse with cell membranes without compromising the integrity of epithelial barriers. Cellfectin II has been utilised for *in vivo* mosquito transfection in various studies (e.g., Cheng 2011, doi:10.1371/journal.pone.0022786), including our own (McFarlane 2021, doi:10.1111/imb.12700). These studies have shown that Cellfectin does not adversely affect mosquito survival, strongly suggesting that it does not impair tissue integrity in mosquitoes. Taken together, it is unlikely that Cellfectin exposure six days before infection could account for the recovered phenotype.

Minor comments:

1. The Fig 1 finding that primarily arthropod-transmitted bunyaviruses encode NSm is intriguing. It would be good for the authors to speculate in the discussion why Bandavirus and Banyangvirus do not appear to encode an NSm protein.

(R): We agree that the convergent evolution of arthropod-transmitted bunyaviruses, with independent and repeated acquisitions of NSm, is indeed intriguing, especially considering the few exceptions among tick-borne bunyaviruses (e.g., Bandavirus and Banyangvirus). The lack of comprehensive knowledge about how bunyaviruses infect and disseminate in ticks *in vivo* complicates the formulation of data-driven hypotheses to explain why insect-borne bunyaviruses evolved to possess NSm while tick-borne bunyaviruses did not. However, several possibilities can be considered. For instance, tick-borne bunyaviruses may not require cell-to-cell spread in the tick gut for propagation and dissemination. Interestingly, while hematophagous insects digest blood in their gut lumen, in ticks, blood digestion occurs intracellularly. This intracellular absorption of blood - and thus virus particles - into gut cells could lead to a higher number of cells being initially infected, resulting in a productive gut infection without the need for viral cell-to-cell spread. Additionally, biological differences such as the prolonged tick blood-feeding process and significant, long-lasting gut distension might explain why tick-borne viruses can disseminate without NSm. The discussion has been revised accordingly (L451-466).

2. Also related to Fig 1: Are there any sequence similarities between the NSm proteins encoded by the divergent bunyaviruses? It would be helpful for the authors to comment on this when discussing Fig. 1 or even provide an alignment in the supplementary figures.

(R): We thank the reviewer for this suggestion. We have blasted the BUNV NSm sequence to retrieve all homologous sequences, aligned them, and performed a

phylogenetic analysis (new Fig. S1a). NSm from most of *Peribunyaviridae* family members could be retrieved and aligned, although they are globally very dissimilar (distance mean: 0.8539). This supports a single ancestral acquisition of a gene – further named NSm - within the *Peribunyaviridae*, followed by divergent evolution in the different viruses of this family. However, NSm from other families, i.e., *Nairoviridae*, *Phenuiviridae* and *Tospoviridae*, could not be retrieved due to an absence of protein sequence similarity. This is also confirmed by the lack of similarity in the structural predictions of the NSm protein across different families (BUNV, RVFV, CCHFV and TSWV), as shown in the new Fig. S1b. Consistent with the law of parsimony (previously discussed in Fig. 1), this would support independent acquisitions of different genes – all called NSm - by the different virus families. This is now presented along with Fig. 1 (L113-124).

It is outside the scope of this manuscript, but it would be interesting to see if trans expression of NSm from a different bunyavirus could also rescue BUN.

(R): Indeed, it would be interesting in a future study to test for complementation of BUN-delNSm cell-to-cell spread using heterologous NSm proteins. Considering our new sequence analysis (Fig. S1) suggesting a unique acquisition event in the *Peribunyaviridae* family, and independent acquisition event(s) in other families, trans expression may work between viruses of the *Peribunyaviridae* family (although sequences are still very dissimilar) but may not work between bunyavirus families. Since different proteins can be involved in a same cell-to-cell spread mechanism and different cell-to-cell spread mechanisms exist, a lack of rescue would not exclude that those different independently acquired genes – now all called NSm – still exert a similar function in cell-to-cell spread. We have modified the discussion to elaborate on this idea (L468-477).

3. Can the authors clarify the bloodmeal doses chosen for these studies. Fig. 2 uses 4×10^8 pfu, Fig 3 uses 5×10^4 , etc. How were the doses selected for each route/experiment?

(R): In Fig. 3, the virus is inoculated by intrathoracic injection to bypass the gut, while in other experiments, mosquitoes are infected orally with an infectious blood meal. It is quite common in the field to orally infect mosquitoes with a dose above 10^7 PFU/mL of blood. When injecting a virus in the body cavity, dose is generally inferior to what would be given orally through blood meal, generally between 10^3 and 10^5 PFU/mosquito depending on publications. Considering that the BUNV-wt titres we measured in bodies following an oral infection were around 10^4 - 10^5 PFU/body, we chose to inject each mosquito with 5×10^4 PFU. This is now clarified in Material and Methods (L583-584).

4. The doses used in Fig 6 to address overcoming the midgut barrier are only 10-fold different and thus make it hard to definitively conclude that dose has not impact on dissemination.

(R): We understand the reviewer's point, and this is a very interesting question in the mosquito-arbovirus field that warrants further investigation. If the hypothesis of "passive viral escape facilitated by transient basal lamina (BL) leakiness" is favoured over an active dissemination mechanism, then a minimal number of infected midgut cells would be required to reach the threshold probability for an infected cell to overlap with the relatively few compromised regions of the BL, resulting in "passive" virus dissemination (originally discussed L385-398). Given the number of mosquitoes analysed, one would expect to observe more frequent dissemination at higher infection doses compared to lower doses. However, it is possible that even at the higher dose tested, the number of infected cells might still be too low to reach this probability threshold. Therefore, we have edited the text to be less affirmative regarding the conclusion on dissemination (L306-309 and L407).

5. Should line 204 read to lower levels for DeINSm compared to wt (phrase reversed)?
(R): We thank the reviewer for spotting this mistake. This has been corrected (L225).

Line 119: It would be helpful to describe in text (outside of methods) and in figure/legend which species and organ these cell lines
(R): Information is now given in the text (L133-134) and in figure legend (Fig. S2). References have been added in methods (L501-502).

6. Figure 4 legend: define the LOD for qRT-PCR. Also define in methods
(R): LoD is a probabilistic measurement of the lowest amount of a target that an assay can detect 95% of the time. Unfortunately, we do not have enough replicates for each concentration in the standard curve (which should be over 20) to properly determine the LOD of our qPCR assay. Therefore, we now present these qPCR data analysed with the delta delta CT method, which does not change the initial conclusion/interpretation of the data, *i.e.*, relative comparison between both viruses and overtime (Fig. 4c).

7. Figure S2: please include the statistical method used to determine no significant difference in S2A/B.
(R): Statistical significance shown on the graph was obtained using a two-tailed Mann-Whitney test. This is now added in figure legend (Fig.S3).

8. Multiple minor typos and discrepancies in how acronyms or measurements are written in text throughout (*i.e.* ml vs mL).
(R): We carefully checked the whole manuscript and corrected all the typos we could spot.

Reviewer #2 (Remarks to the Author):

I co-reviewed this manuscript with one of the reviewers who provided the listed reports.

This is part of the Nature Communications initiative to facilitate training in peer review and to provide appropriate recognition for Early Career Researchers who co-review manuscripts.

Reviewer #3 (Remarks to the Author):

The manuscript titled “NSm is a critical determinant for bunyavirus transmission between vertebrate and arthropod hosts” by Terhzaz S. et al. describes that the Bunyamwera virus (BUNV) nonstructural protein NSm is dispensable for viral replication in mosquito cell culture, but necessary for viral replication and spread in the midgut cells of *Aedes aegypti* mosquitoes. Meanwhile, intrathoracic injection of BUNV lacking the NSm gene resulted in efficient viral spread, indicating a specific function of NSm in the midgut cells. The authors also showed that (i) the in trans expression of NSm facilitates the replication of BUNV lacking the NSm gene, and (ii) increasing the dose of BUNV lacking NSm does not lead to viral spread in the midgut. Imaging of NSm protein in the mosquito midgut epithelium was also performed. The manuscript is generally well-written, and this study will provide new insight into the function of the orthobunyavirus NSm protein in mosquito vectors. However, apparently, the interpretation of the experimental result beyond the orthobunyaviruses has not been validated. For example, in the case of Rift Valley fever virus (RVFV), it is the 78kD protein, not the NSm protein, that has been shown to be responsible for dissemination via the mosquito midgut. In terms of technical aspects, it is important to include the threshold of detection for plaque assays and to ensure the inclusion of mock-infected controls for the validation of immunological detection of viral antigens. Specific points are as follows:

(R): We thank the reviewer for their useful feedback and provide point-by-point answers below.

Major points

- Lines 101 – 110 and Figure 1: The authors generalized the presence of NSm proteins across arboviruses, including the Phenuiviridae. However, for instance, RVFV encodes a 78kD protein for viral dissemination via the midgut of *Aedes aegypti* mosquitoes, whereas specific deletion of NSm did not impair viral dissemination via the midgut (Kreher et al., *Emerg. Microbes Infect.* 2014, 3: e71). Meanwhile, the presence of 78kD-like proteins in other phleboviruses is poorly characterized. Given this complexity, direct comparison of phenuiviridae NSm with that of BUNV or other bunyaviruses in this study seems challenging.

(R): In Figure 1, we analysed the presence or absence of NSm in the genomes of bunyaviruses, demonstrating that NSm is exclusively present in arboviruses. While RVFV indeed encodes P78, which consists of NSm-Gn, RVFV also encodes NSm (P14) separately; both NSm and P78 are expressed/translated from different start

codons. We have added a note regarding this point in the figure legend (Fig. 1). Nevertheless, we acknowledge that the role of NSm in RVFV-infected mosquitoes appears to be more complex than in BUNV. We apologise for overlooking the Kreher *et al.* paper, which shows that the specific knockout of P78 – and therefore the NSm version fused to Gn - affects mosquito dissemination while the P14 NSm knockout does not. This seems to contrast with the earlier conclusion by Crabtree's paper, which suggested that P14 NSm is required for dissemination in mosquitoes. However, the NSm mutation used in Crabtree's paper was in fact also disrupting P78 and therefore the whole NSm-Gn, reconciling the two studies. The discussion has been modified to clarify this point (L431-432). As described in the Kreher *et al.* paper, RVFV P14 NSm remains on the cytosolic side of the ER membrane, while in P78 (NSm-Gn), NSm acts as a transmembrane protein entering the lumen of the ER, like BUNV NSm. It is thus possible that the exposition of NSm on the lumen of the ER (and therefore on the exterior of cells), as in RVFV P78 and BUNV NSm, is required for NSm role in virus cell-to-cell spread and dissemination in mosquitoes.

- Similar to the comment above, insect-specific bunyaviruses like Gouleako virus do not encode the NSm protein. Further elaboration on the potential distinctions between arboviruses and insect-specific bunyaviruses regarding virus transmission would strengthen the rationale of this study.

(R): This is indeed an interesting question. The knowledge about insect-specific viruses (ISVs) is still limited, and some aspects need to be clarified, such as the mechanisms of transmission, tissue tropism and persistence in mosquitoes. Therefore, it is difficult to elaborate data-driven hypotheses to explain why arboviruses encode an NSm protein while ISVs do not. A major difference between arboviruses and ISVs in their mode of acquisition and transmission may explain why ISVs do not encode an NSm protein. Arboviruses are acquired through a blood meal and need to move efficiently between cells in the midgut and then disseminate to other parts of the mosquito body. For ISVs, which may not face the same transmission and barrier challenges, the NSm protein might be unnecessary. These viruses could have evolved different mechanisms suited to their acquisition modes, such as vertical or venereal transmission, where direct access to mosquito tissues, and their maintenance in mosquitoes, bypasses the midgut challenges faced by arboviruses. This is now discussed (L441-449).

- Figure 3e, f, g: negative controls (mock-infected) should be included to validate the detection of the virus N antigen.

(R): We have now included in Supplementary figure S4a an anti-N immunostaining on midguts and salivary glands from non-infected mosquitoes (e.g., no expression of N) validating the anti-N immunostaining assay on midguts.

• Figure 2b-c, 3b-d, 4b, 5f, 6b, d, and S2a-b: In the plaque assay, is it possible to detect 1 PFU/ml (i.e., 10^0 PFU/ml)? If not, please specify the limit of detection in each graph.
(R): Mosquito are sampled and homogenised in small volumes (e.g., 100 μ L) and therefore it is not possible to measure undiluted samples in our conditions (12 well plates). The detection limit is therefore 50 PFU/mL (i.e., $10^{1.69}$ PFU/mL). It is now added in Material and Methods (L618-619).

• Figure 5. Mock-transfected cells should be stained with anti-V5 to validate the immunostaining assay using midguts. In addition, authors should describe how intrathoracic injection of transfection mixture can reach to midgut epithelium: e.g., crossing basal lamina?

(R): We have now included in Supplementary figure S4b an anti-V5 immunostaining on midguts from mosquitoes transfected with the plasmid pPUB-Gal4 (e.g., no expression of V5), validating the anti-V5 immunostaining assay on midguts.

While it is known in various models that molecules can be selectively transported across the basal lamina, the mechanistic principles behind this selective permeability barrier are still not fully understood. To our knowledge, it remains unclear how Cellfectin/DNA can cross the basal lamina. It is worth mentioning that we are not the first to demonstrate that *in vivo* transfection facilitates the expression of DNA in the midgut of mosquitoes. (see Figure S1 in Cheng G, Liu L, Wang P, Zhang Y, Zhao YO, et al. (2011) An In Vivo Transfection Approach Elucidates a Role for Aedes aegypti Thioester-Containing Proteins in Flaviviral Infection. PLoS ONE 6(7): e22786. doi:10.1371/journal.pone.0022786). This reference is now added in Material and Methods (L535).

Minor points

• Lines 89-90, 101 – 110: “arbovirus” should be defined as plant virus is also included in Figure 1.

(R): We have edited the text to mention that arboviruses can also be transmitted by arthropods to plants (L92).

• Please define the “PFU/ml (Mosquito)” in graphs.

(R): The titre of infectious particles is expressed as a concentration in PFU/mL for each sample (whole mosquitoes, tissues or saliva). The absolute quantity in each sample is 10 times less for all samples (sampled in 0.1 mL) and 20 time less for saliva samples (sampled in 0.05 mL). This has now been defined in Material and Methods (L619-620).

• Line 200: “Interestingly, in contrast to virus titres, the levels of viral segment S levels were similar for both viruses at 3 and 24h, which suggests that viral RNA from initially

ingested virions persists in the midgut for 24h post-blood meal, likely in the gut lumen while the blood is not yet fully digested.” This does not fully account for the absence of infectious virus at 24 hpbm.

(R): We did not mean here to explain the absence of infectious virus at 24h pbm, but we just wanted to highlight the persistence of viral RNA traces while there is not infectious particle anymore.

- Line 297: “BUNV-NSmV5” should be explained further in terms of the location of V5-tag insertion.

(R): BUNV-NSmV5 contains the V5 epitope inserted between residues 403 and 420 of the NSm coding region. This plasmid was described previously and is referenced in the manuscript. The V5-tag location has been added in Material and Methods (L521-522).

- Line 404-405: Please specify “RVFV lacking both 78kD and NSm genes”.

(R): This has been clarified (L431-432).

- Line 407: “Thus, the critical function of NSm in cell-to-cell spread may be a conserved feature of different bunyavirus families.” By ignoring the 78kD function, this description is not very logical.

(R): As mentioned above, the text has been modified to clarify this point.

- Figure S1a: The illustration of BUN M requires further clarification in terms of the deletion site of NSm region: e.g., $\Delta 303 - 331$?

(R): This has been clarified in the figure legend (new Fig. S2a).

REVIEWER COMMENTS

We sincerely thank the reviewers for their feedback on our revised manuscript. We have addressed each comment and provided a detailed, point-by-point response (R) below. The manuscript has been revised accordingly.

Reviewer #3 (Remarks to the Author):

The revised manuscript titled "NSm is a critical determinant for bunyavirus transmission between vertebrate and arthropod hosts" by Terhzaz S. et al. has addressed most of the reviewers' concerns. However, two minor issues remain that should be addressed to maintain scientific accuracy.

Specific points:

1. Figure 1 legend: The statement "Phleboviruses (e.g., Rift Valley Fever virus) also encode P78, which consists of an NSm-Gn fusion protein" is not scientifically accurate. The 78kD protein is not identical to the NSm-Gn fusion protein, as it can also be expressed separately from NSm or the 78kD protein. The 78kD protein contains a signal peptide and is likely structurally distinct from the NSm protein or the NSm-Gn protein.

(R): We did not intend to distinguish P78 from an NSm-Gn protein lacking a signal peptide. Our goal is to emphasize that in Phleboviruses, the M segment encodes both NSm (P14) and P78, with the NSm sequence positioned upstream of the Gn region. As noted in our discussion, research using various RVFV mutants has shown that dissemination from the gut relies on P78 rather than NSm (P14). In our point-by-point response to the initial reviews, we acknowledged that the structure and cellular localization of NSm in these different proteins (P78 with or without a signal peptide, P14) could influence its function, which might explain why P14 is not essential for dissemination, whereas P78 is. For the sake of clarity, we opted not to include these intricate details in the manuscript discussion, as we believe they are unnecessary for the general audience of Nature Communications.

To ensure scientific accuracy, we have revised the Figure 1 legend to: "*Phleboviruses (e.g., Rift Valley Fever virus) also encode P78, which is processed from the glycoprotein precursor initiated from the first start codon of the M segment, encompassing both the NSm and Gn coding regions.*"

2. Figures 2b-c, 3d, 4b, 5f-i, 6b, d, and S3a-b: The Y-axis should include a threshold line, and undetected samples should be plotted as "< 50 PFU/ml" or "undetectable." This reviewer believe that the plotting these onto 10^0 is not scientifically accurate.

(R): While it is not entirely accurate to plot negative samples (undetectable samples with no plaque at the first dilution, below the detection limit of $10^{1.69}$ PFU/mL) at 10^0 , this choice is arbitrary and commonly accepted in the virology field to present data in a clearer manner. Plotting these samples on the threshold line or between the threshold line and 10^0 is also misleading, as their true values remain unknown due to being below the detection limit of our technique. To clarify this point, we have revised the Methods section as follows:

“Since samples could not be measured without dilution, detection limit is 50 PFU/mL (i.e., $10^{1.69}$ PFU/mL). To present titres including negative ones in log scale, +1 was added to all values including the negative ones (i.e., $1=10^0$). Undetectable samples (no plaque at the first dilution, below the detection limit $10^{1.69}$ PFU/mL) are plotted at 10^0 PFU/mL.”